# Discovery of deep-sea coral symbionts from a novel clade of marine bacteria with severely reduced genomes

Samuel A. Vohsen [1,2,3], Harald R. Gruber-Vodicka[4,5], Santiago Herrera [2,3], Nicole Dubilier [4], Charles R. Fisher[1] & Iliana B. Baums [1,6,7,8] ✉

Microbes perform critical functions in corals, yet most knowledge is derived from the photic zone. Here, we discover two mollicutes that dominate the microbiome of the deep-sea octocoral, *Callogorgia delta*, and likely reside in the mesoglea. These symbionts are abundant across the host's range, absent in the water, and appear to be rare in sediments. Unlike other mollicutes, they lack all known fermentative capabilities, including glycolysis, and can only generate energy from arginine provided by the coral host. Their genomes feature several mechanisms to interact with foreign DNA, including extensive CRISPR arrays and restriction-modification systems, which may indicate their role in symbiosis. We propose the novel family Oceanoplasmataceae which includes these symbionts and others associated with five marine invertebrate phyla. Its exceptionally broad host range suggests that the diversity of this enigmatic family remains largely undiscovered. Oceanoplasmataceae genomes are the most highly reduced among mollicutes, providing new insight into their reductive evolution and the roles of coral symbionts.

Corals are foundation species that support diverse animal communities from shallow waters to the deep sea. Corals also associate with a diversity of microbes[1], including the well-studied algal symbionts of the family Symbiodiniaceae and other microbial taxa that perform a variety of roles, from providing nitrogen through fixation[2] to causing disease[3]. Most of these microbes were identified in scleractinian corals from the photic zone[1], while studies investigating the roles of microbes in octocorals[4] and deep-sea corals are rarer because of the limitations that great depths impose on sampling and experimentation. Such work requires the use of remotely operated vehicles or submersibles launched from ships. Still, studies have demonstrated that octocorals and deep-sea corals host some of the same associates as shallow-water scleractinians, such as corallicolid apicomplexans[5–8] and *Endozoicomonas*[9–12].

However, one study comparing deep-sea coral microbiomes to those in shallow-water corals found differences in metabolic activities, including increased anaerobic ammonia oxidation and increased chitin degradation[13]. In addition, deep-sea octocorals also host associates that are rare or absent in shallow-water and/or scleractinian corals. These associates include bacteria from the SUP05 cluster, whose role is linked to cold seeps in the deep sea[14], and members of the class Mollicutes.

Members of the class Mollicutes associate with many coral species and a wide diversity of plant, fungal, and animal hosts, including humans[15,16]. Several members are well-studied parasites[17], while the impact of others on their hosts is unclear such as the ubiquitous intracellular symbionts of arbuscular mycorrhizal fungi[18–20]. Yet others are mutualists, such as *Spiroplasma* spp., that infect insects and confer

[1]Department of Biology, The Pennsylvania State University, State College, PA, USA. [2]Department of Biological Sciences, Lehigh University, Bethlehem, PA, USA. [3]Lehigh Oceans Research Center, Lehigh University, Bethlehem, PA, USA. [4]Department of Symbiosis, Max Planck Institute for Marine Microbiology, Bremen, Bremen, Germany. [5]Zoological Institute, Christian-Albrecht University of Kiel, Kiel, Schleswig-Holstein, Germany. [6]Helmholtz Institute for Functional Marine Biodiversity (HIFMB), Carl von Ossietzky University of Oldenburg, Oldenburg, Lower Saxony, Germany. [7]Alfred-Wegener-Institute, Helmholtz-Centre for Polar and Marine Research (AWI), Bremerhaven, Bremen, Germany. [8]Institute for Chemistry and Biology of the Marine Environment (ICBM), School of Mathematics and Science, Carl von Ossietzky University of Oldenburg, Oldenburg, Lower Saxony, Germany. ✉e-mail: iliana.baums@hifmb.de

protection against nematodes, parasitoid wasps, and fungi through the production of unique toxins[21]. Reflecting their lifestyles as symbionts, their evolutionary history is dominated by genome reduction, with some species considered to have the most reduced genomes capable of supporting cellular life[22,23]. Recently, novel and divergent mollicutes were discovered in diverse marine invertebrates, including a deposit-feeding holothurian from a deep-sea trench[24]; pelagic, photosynthetic jellyfish[25,26]; wood-boring, deep-sea chitons[27]; ascidians from a coastal lagoon[28], and crown-of-thorns sea stars from the Great Barrier Reef[29]. However, the impacts these symbionts have on their hosts remain unclear.

Among coral species, mollicutes are most commonly found in octocorals[30–39], but have also been reported in black corals[40] and stony corals[41,42]. These corals occupy a wide breadth of habitats from shallow-water reefs[36,39,43], through the mesophotic zone[40] and the deep sea[31,32,41] to abyssal depths[44]. In these coral species, *Mycoplasma* spp. are the dominant microbes, comprising more than 50% of their associated microbial communities[32,39,43]. Interestingly, the relative abundances of *Mycoplasma* vary substantially across the ranges of individual coral species[38,43]. Thus, this association may be influenced by environmental conditions or otherwise shaped by geography, such as through limited dispersal. In addition, *Mycoplasma* spp. exhibit varying degrees of host specificity[33,38,39]. Co-occurring coral species host different *Mycoplasma* variants[33] suggesting that they may have adapted to their coral hosts and closely interact with them as symbionts. Despite these insights, the interactions between mollicutes and their host corals remain unknown.

Here, we discover two novel bacteria of the class Mollicutes, which are abundant in the deep-sea octocoral *Callogorgia delta*. *C. delta* is common along the continental slope in the Gulf of Mexico between 400 and 900 m depth, where it is often the dominant habitat-forming coral species[45,46]. Like many deep-sea coral species, *C. delta* colonies create habitats for numerous animal species, including *Asteroschema* ophiuroids and the chain catshark, *Scyliorhinus retifer*, which lay their eggs directly on *C. delta* colonies[47].

To characterize the association between *Callogorgia delta* and these newly discovered mollicutes, we screen colonies from six sites in the Gulf of Mexico across 3 years of sampling (Fig. 1a) using 16S metabarcoding to determine the prevalence of these novel mollicutes and quantify their relative abundances. We also screen the closely related species *Callogorgia americana* to assess the phylogenetic breadth of the association. *C. americana*, is another dominant habitat-forming species in the northern Gulf of Mexico and provides a good comparison species. *C. americana* occurs at shallower depths (300–400 m) while *C. delta* is often found near hydrocarbon seeps[46]. Further, we assess the specificity of the association to corals by screening sediment and water in addition to *Callogorgia*. We assemble the genomes of these novel mollicutes to describe their metabolic capabilities, determine their phylogenetic positions, and compare them to other mollicutes. To complement metabolic inferences, we sequence metatranscriptomes and so identify active genes and those with the highest transcription levels. Finally, we locate these mollicutes within coral tissue using catalyzed reporter deposition fluorescence in situ hybridization (CARD-FISH) microscopy and transmission electron microscopy.

## Results
### Occurrence of novel mollicutes in and around *Callogorgia*
We investigated the presence of novel mollicutes in *Callogorgia* colonies using V1-V2 16S rRNA-based metabarcoding on an Illumina MiSeq. Colonies were collected from seven locations (Fig. 1a) across 4 years (2010, 2015–2017). Three abundant amplicon sequence variants (ASVs) were identified among 108 *Callogorgia delta* colonies that belong to the class Mollicutes (Molli-1, 2, 3). Molli-1 was the most prevalent ASV (Fig. 1b). It was detected in 99 out of 108 colonies and was present in

samples from all six sampling locations as well as every year each site was sampled (Fig. 1b). Molli-1 had an average relative abundance of 77% among corals which harbored it (determined from frozen samples processed with DNeasy powersoil kits) and reached a maximum of 99% of the microbial community in some samples. It was also present in all four *Callogorgia americana* colonies averaging 93% of the microbial community. Molli-1 constituted a high proportion of the microbiome in most samples including those from 2015 which were preserved in ethanol as well as *C. americana* samples which were processed using a DNeasy Allprep kit (Fig. 1b).

Molli-2 differed from Molli-1 by a single base pair among 300. This closely related ASV was detected in four out of ten colonies from the deepest site, GC290. Among those four colonies, Molli-2 averaged 90% (FPS) of the microbial community while Molli-1 was undetected (Fig. 1b).

Molli-3, which was 72% identical to Molli-1, was prevalent and abundant in some colonies, but less so compared to Molli-1 (Fig. 1b). Molli-3 was present in 69 out of 108 *C. delta* colonies (mean 14.5% when present, up to 82% of the community, FPS) and two out of four *C. americana* colonies (up to 15%). However, Molli-3 was absent in all *C. delta* colonies from the deepest sites GC249 ($n$ = 14) and GC290 ($n$ = 10).

None of these ASVs were detected in any water sample ($n$ = 30). However, Molli-1 and Molli-3 were detected in 35 and 11 out of 44 total sediment samples with maximum relative abundances of 1.1 and 0.16%, respectively and always fewer than 40 reads. Molli-1 was detected in three of four sequencing blanks (0–5 reads).

In addition to novel mollicutes, the most abundant ASVs among *Callogorgia delta* colonies were classified as Thiobarbaceae (mean 7.5% per colony, FPS), *Endozoicomonas* (1.9%), Thioglobaceae (1.5%, most classified as SUP05 cluster), and *Shewanella* (1.2%) (Fig. S1). ASVs with other classifications averaged a total of 6.6% and included other common coral associates such as corallicolid apicomplexans and Rickettsiaceae.

### Phylogenetic positions of the mollicute in *Callogorgia delta*
Twenty-two full-length 16S rRNA gene sequences corresponding to Molli-1 were assembled from separate *C. delta* metagenomes, and three additional sequences were assembled corresponding to Molli-3. These sequences were distinct from those of other known mollicutes and clustered with sequences from recently discovered associates of marine invertebrates including ascidians, jellyfish, sea stars, sea cucumbers, and chitons (UFBoot 100%, Fig. 2a). Interestingly, the two novel mollicutes in *Callogorgia delta* are not each other's closest relative and shared only 84% sequence identity across the entire 16S rRNA gene which is below the suggested cutoff to differentiate bacterial families[48].

In addition, related sequences (≥90% identical) were identified from publicly available 16S rRNA amplicon datasets using IMNGS[49]. Sponges (*Cinachyrella* and *Suberites*) and jellyfish (*Mastigias*) from marine lakes and open water in Indonesia hosted bacterial communities comprised of up to 86% of these related sequences.

A circular and closed genome corresponding to Molli-1 was assembled from a deeply sequenced metagenome of *Callogorgia delta*. The metagenome was filtered to kmers with a depth of 1000 or greater, and the corresponding reads were subsampled at a rate of 10% and assembled with SPAdes to produce a single circular scaffold. In addition, a genome corresponding to Molli-3 was obtained by co-assembling the two metagenomes with the highest 16S coverage and binning contigs using coverages from all 26 metagenomes. Phylogenomic analysis of shared amino acid sequences among mollicute genomes reiterated phylogenetic inferences based on 16S rRNA (Fig. 2b). A novel clade of Mollicutes that associates with marine invertebrates was well-supported (UFBoot 100%) and this clade clustered alongside the intracellular symbionts of mycorrhizal fungi with high confidence (UFBoot ≥98%).

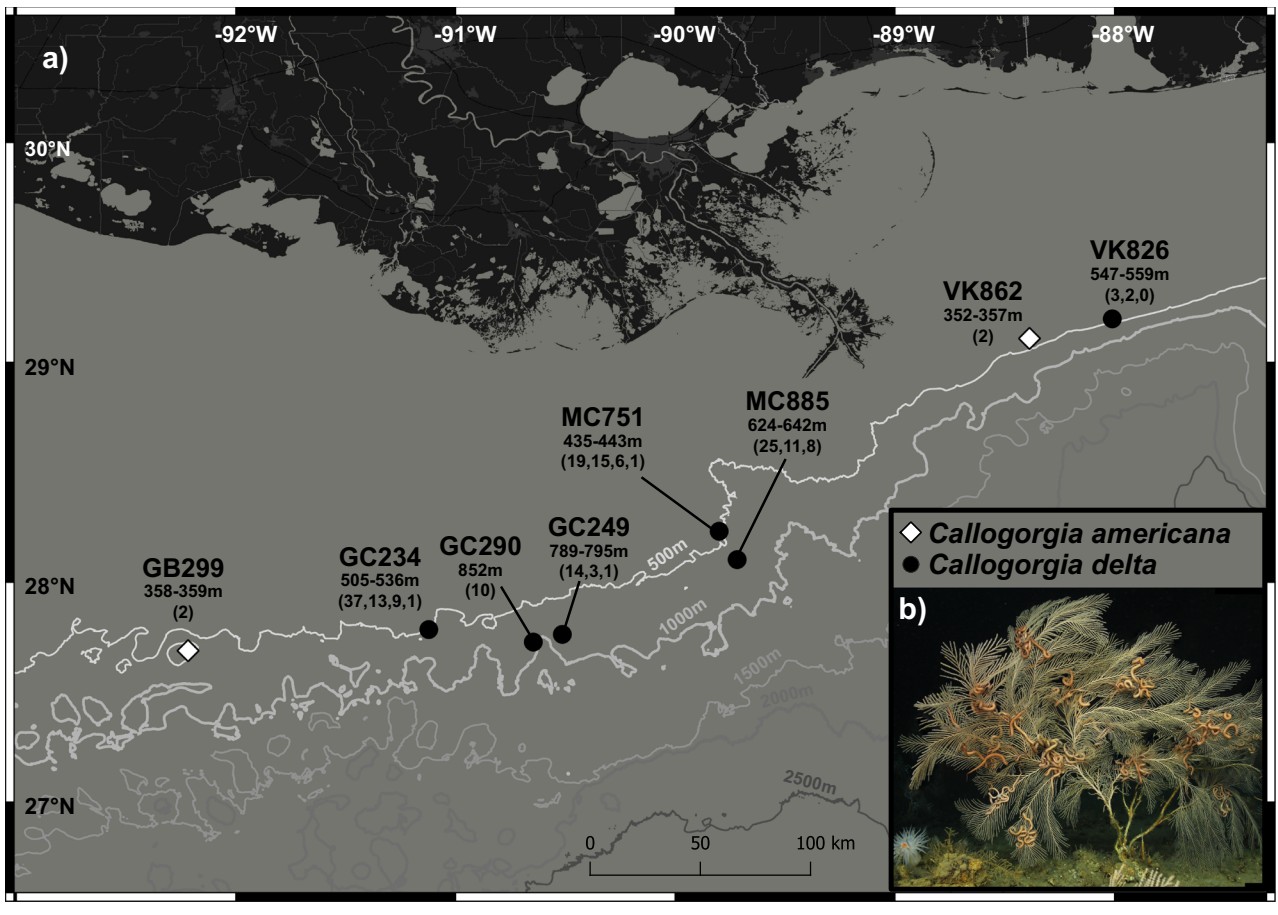

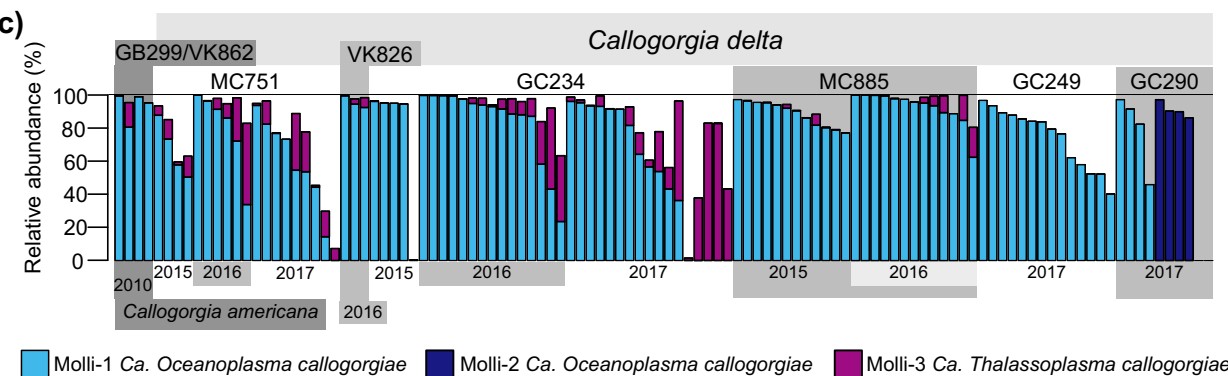

**Fig. 1 | Sampling locations and prevalence of novel mollicutes in *Callogorgia delta* and *C. americana*. a** Map of locations where *Callogorgia delta* (black circles) and *C. americana* (gray diamonds) were sampled. Lines represent 500 m isobaths. The depth range of samples collected from each site is denoted below the site name followed by the number of colonies, sediment samples, niskin water samples (~2.5 L), then McLane pump water samples (~400 L) in parentheses. If no water or sediment samples were collected, only the number of colonies is shown. If no McLane pump water samples were taken, only the number of niskin water samples are shown. Map made using QGIS. Basemap attributions [World Light Gray Base]: ESRI, DeLorme, HERE, MapmyIndia. Bathymetric contours obtained from the Bureau of Ocean Energy Management. **b** Image of *Callogorgia delta*. **c** The relative abundances of novel mollicute ASVs. Each column represents the microbial composition based on 16S rRNA amplicon libraries obtained from a single colony. Colonies are organized by species, site, and sampling year. *C. delta* sites are ordered left to right by increasing depth. *C. americana* samples were extracted with a DNeasy allprep kit and *C. delta* samples from 2015 were preserved in ethanol. All other samples shown were frozen and extracted with DNeasy powersoil kits. For colonies with multiple replicates, the first replicate using frozen tissue is shown, and others are excluded.

Branch lengths between members of the novel clade of marine mollicutes were long. The average nucleotide identities (ANI) between the four representative genomes ranged from 63 to 65% (OrthoANI) suggesting that both mollicutes described here are novel genera. We propose the names *Candidatus Oceanoplasma callogorgiae* gen. nov. sp. nov. and *Ca. Thalassoplasma callogorgiae* gen. nov. sp. nov. corresponding to Molli-1 and Molli-3, respectively. Further, we propose the novel family Oceanoplasmataceae with *Ca. Oceanoplasma callogorgiae*

as its type species and which includes other associates of marine invertebrates (etymologies and taxon descriptions can be found at the end of the Results section).

## Metabolic capabilities of the novel mollicutes in *Callogorgia delta*

The genome of *Ca. Oceanoplasma callogorgiae* was highly reduced in size (446,099 bp, Table 1 and Fig. 2c) and function with only 359

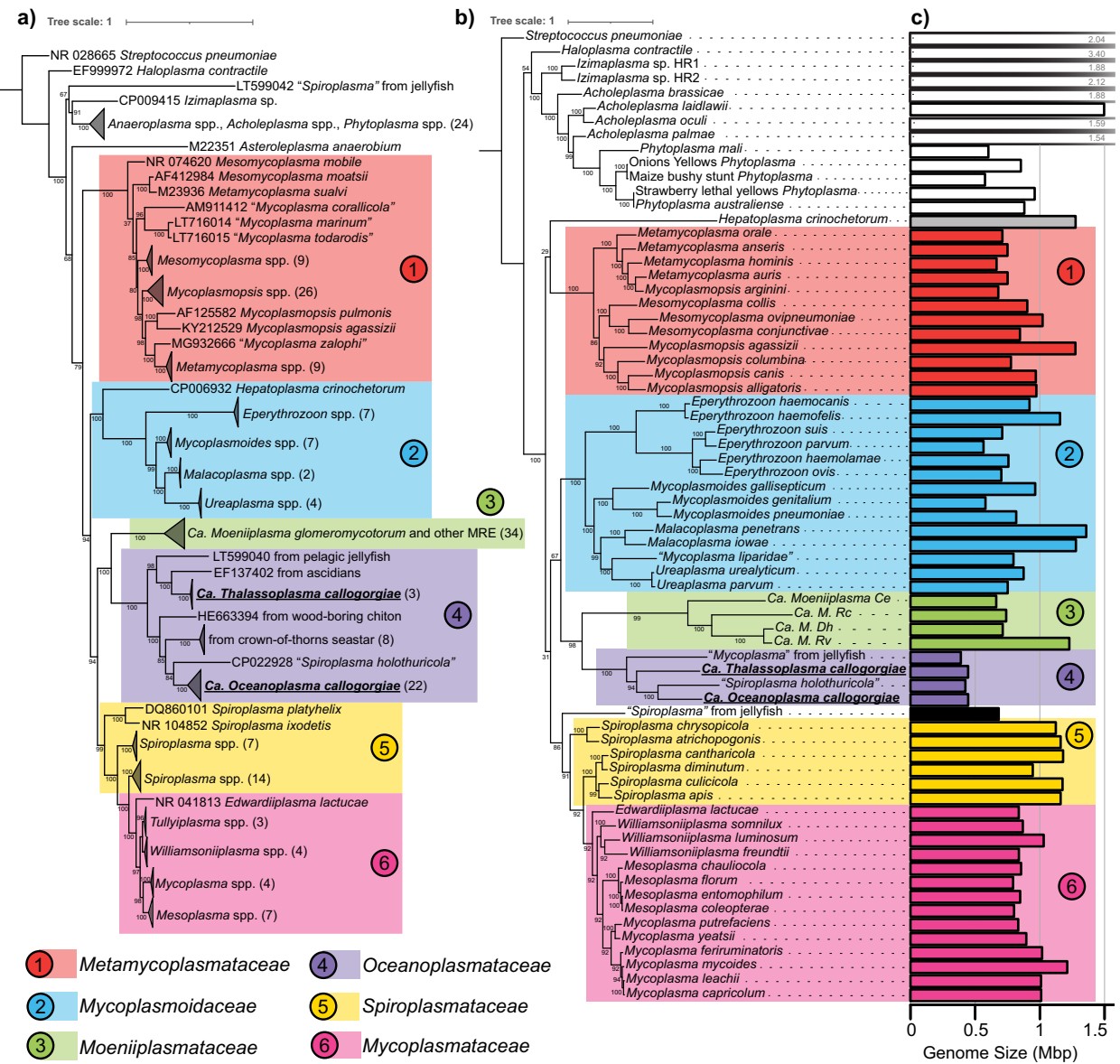

**Fig. 2 | Phylogenetic analyses of the novel mollicutes in *C. delta*.** Maximum likelihood phylogenetic trees of class Mollicutes using the (**a**) 16S rRNA gene and (**b**) amino acid sequences of 43 genes. UFBoot support values are reported at each node. Numbers in parentheses designate the number of sequences within collapsed nodes. *Ca. Oceanoplasma callogorgiae* and *Ca. Thalassoplasma callogorgiae* are presented in bold text and underlined. Genome sizes (in Mbp) are depicted in (**c**).

predicted protein-encoding genes (241 with KEGG orthology annotations). The genome lacked any pathway to generate biomass or energy from carbohydrates and instead contained the genes comprising the arginine dihydrolase pathway (*arcABCD*) and ATP synthase. Genes from this pathway were represented by transcripts in both libraries, were among the top fifteen genes with the highest transcription levels (mean transcripts per million (tpm) 9300–96,900), and constitute the sole pathway in this genome capable of producing ATP (Fig. 3). Statistical analyses were not conducted to support the ranking of transcription levels due to the low sample size ($n = 2$). This pathway generates ATP from the equimolar catabolism of arginine. In other non-fermentative mollicutes, additional ATP is generated as the resulting ammonia establishes a proton gradient which drives ATP synthase[50,51]. Besides arginine catabolism, a few genes could potentially be involved in ATP production: glyceraldehyde-3-phosphate dehydrogenase (*gapA*), phosphoglucomutase (*pgm*), ribose-phosphate pyrophosphokinase (*prsA*), and acetate kinase (*ackA*).

The genome featured a very limited ability to synthesize essential compounds. It encoded only one enzyme involved in amino acid synthesis (threonine dehydrogenase) but several protein-degrading enzymes and amino acid importers. Similarly, it encoded no enzymes that synthesize vitamins, cofactors, or coenzymes besides NAD⁺ as well as FMN and FAD from imported riboflavin (Fig. 3a).

The genome contained extensive and active mechanisms to interact with foreign DNA, including type I, II, and III restriction-modification systems and a Class 2 Type II-C CRISPR-Cas system. The specificity subunit S of its type I restriction-modification system had the 4th highest average transcription level among all genes (mean tpm = 87,800, Fig. 3b). Further, the genome contained the second highest number of spacer sequences, at 107, of all available genomes from mollicutes[52,53]. The potential targets of these spacers could not be identified. However, CRISPRCasdb[54] identified three spacers that partially resembled those found in the genomes of other marine bacteria (*Empedobacter falsenii*, *Sulfodiicoccus acidiphilus*, and *Nostoc sphaeroides*: all alignment lengths 17/30 bp, all blast *e*-values 0.007).

**Table 1 | Summary of the genome statistics for the novel mollicutes in *Callogorgia delta* and their closest known relatives**

| Organism name | Genome size (bp) | Contigs | GC (%) | DRAM Pegs (w/ KO) | Classic RAST CDS | tRNAs | rRNAs | checkm Completeness (%) | Contamination (%) | Predicted genes | Coding density (%) |
|---|---|---|---|---|---|---|---|---|---|---|---|
| *Ca. Oceanoplasma callogorgiae* | 446,099 | 3 | 31.0 | 359 (241) | 405 | 34 | 2 + 7* | 70.4 | 2.6 | 392 | 88.2 |
| *Ca. Thalassoplasma callogorgiae* | 445,436 | 21 | 24.9 | 385 (237) | 416 | 30 | 2 | 72.4 | 2.6 | 396 | 87.6 |
| *Ca. Spiroplasma holothuricola*[25] | 424,539[24] | 2[24] | 29.5[24] | 336 (224) | 399 (347[24]) | 32 | 2 | 67.4 | 2.3 | 367 | 89.9 |
| *Mycoplasma*-like from jellyfish[25] | 389,210 (409,158[25]) | 6 (11[25]) | 33.1 (32.9[25]) | 342 (225) | 358 | 32 | 2 | 73.4 (74.1[25]) | 2.3 (2.3[25]) | 356 | 92.5 |

CDS coding sequences, Pegs (w/KO) protein-encoding genes with KEGG orthologies.
*LSU rRNA fragments. References: He et al.[24], Viver et al.[25]

Four predicted genes with no functional annotations were represented by transcripts in both libraries and among the ten genes with the highest transcription levels. (mean tpm = 14,500–98,500, Fig. 3b). The highest of these (named U1) was predicted to contain a transmembrane segment and a signal peptide (Hhblits). It also contained a region that matches the EAL domain of various proteins. The EAL domain is a diguanylate phosphodiesterase, which degrades the second messenger cyclic di-GMP[55–57]. Another of these, unannotated gene (U2), was predicted to contain a GIY-YIG endonuclease domain with high confidence (>95%, Hhblits). The fourth of these genes, U4, belongs to the YneF family, whose function is unknown but is conserved across Mollicutes and most Bacilli where it is highly expressed and considered an essential protein[58].

The genome of *Ca. Thalassoplasma callogorgiae* was very similar to that of *Ca. O. callogorgiae*. It was reduced in size (445,456 bp) and metabolic capabilities with only 385 predicted protein-coding genes (237 with KEGG orthology annotations). It possessed all the pathways and genes mentioned above for *Ca. O. callogorgiae* except it lacked a type II restriction-modification system and homologs of U1 and U2. In contrast, it possessed ribose-5-phosphate isomerase (*rpiB*) and alternative genes to synthesize NAD$^+$ (Fig. 3a).

## Comparison to other mollicutes

The genomes of *Ca. O. callogorgiae* and *Ca. T. callogorgiae* were similar to others within Oceanoplasmataceae. The genome of *Ca. Spiroplasma holothuricola* is publicly available, and we assembled a fourth genome corresponding to the closely related associate of the jellyfish *Cotylorhiza tuberculata*[25]. To do so, we assembled the metagenomic sequence library from that study with the highest coverage of *Mycoplasma* 16S rRNA. We then mapped all four metagenome libraries to this assembly to use for binning with MetaBat 2[59] and identified bin 1 as a member of Oceanoplasmataceae. These four genomes shared 211 clusters of orthologous genes (COGs) which comprised 60–66% of the COGs in each genome. Besides housekeeping genes, all four genomes encode the arginine dihydrolase pathway and CRISPR-Cas systems. The genome of the jellyfish associate was the most divergent and may have further but limited means to generate ATP since it encoded lactate dehydrogenase, glycerate kinase, ribulose-phosphate 3-epimerase, and an agmatine/putrescine antiporter.

Overall, the genomes of Oceanoplasmataceae were distinct from other mollicutes and were the most reduced (Fig. 4). They formed a distinct cluster based on the presence/absence of COGs (Fig. 4a), were the smallest (389–446 kbp versus >564 kbp[60]; Fig. 3c), and had the fewest protein-encoding genes (pegs) (336–385 versus >515; Fig. 4b). Further, they were the only mollicutes missing the genes necessary to conduct glycolysis besides *Moeniiplasma* spp. with which they shared other physiological similarities. Compared to other mollicutes, they had the fewest COGs with KEGG orthology terms and the fewest genes involved in carbohydrate metabolism. Oceanoplasmataceae genomes were distinguishable from those of *Moeniiplasma* spp. by the presence of genes encoding ribose-phosphate pyrophosphokinase, glyceraldehyde-3-phosphate dehydrogenase, and ATP synthase.

All Oceanoplasmataceae genomes contained twenty-three COGs that were absent in all other mollicutes. Notably, these included a Ca$^{2+}$/Na$^+$ antiporter, several other transporters, and a thymidine kinase. Thymidine kinases were enriched in the genomes of Oceanoplasmataceae, each containing four copies, while all other genomes had one or zero except two *Moeniiplasma* spp. which had two copies.

## Localization of bacteria in *Callogorgia delta*

To locate the symbiotic bacteria, we used probes that bind to ribosomal RNA, permitting deposition of a fluorophore through catalyzed

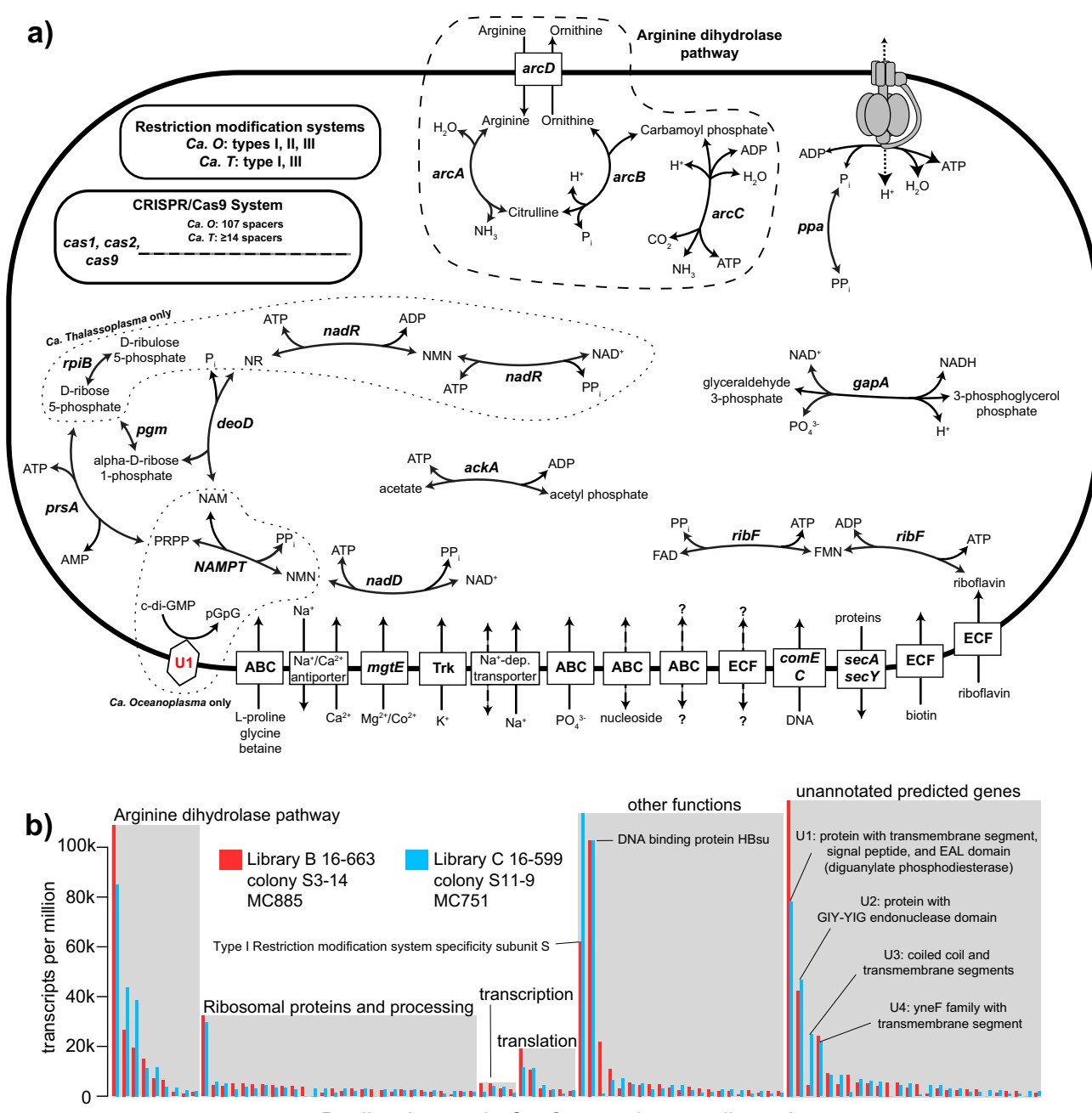

**Fig. 3 | Simplified metabolic model and transcriptomic profile of novel molli-cutes. a** Simplified metabolic model of *Ca. Oceanoplasma callogorgiae* and *Ca. Thalassoplasma callogorgiae*. Gene names are denoted in italics and unidentified proteins in red text. ATP is generated by the arginine dihydrolase pathway. Additional ATP is likely generated by ATP synthase utilizing the proton gradient resulting from the consumption of $H^+$ and production of $NH_3$. Nicotinamide (NAM), nicotinamide riboside (NR), nicotinamide mononucleotide (NMN), 5-phosphoribosyl diphosphate (PRPP). **b** Transcription levels of *Ca. O. callogorgiae* genes from two libraries sorted by functional annotations. Only genes with a maximum of 1500 transcripts per million among both libraries are displayed.

reporter deposition fluorescence in situ hybridization (CARD-FISH). Fluorescence representing abundant bacteria was observed in the mesoglea of four out of five separate *Callogorgia delta* colonies (Figs. 5 and 6). These signals were present in tissue sections hybridized with the EUB338 I-III probe mix but absent in adjacent sections hybridized with the negative control probe (non-EUB). These signals were <1 μm in diameter and overlapped DAPI fluorescence. These bacterial signals were observed in both the mesoglea within the coe-nosarc surrounding the proteinaceous central axis (Fig. 5b–d) as well as the mesoglea within polyps (Fig. 6a–c). They formed large aggre-gates that conformed to the shape of the mesoglea and potentially contained hundreds or thousands of cells (Figs. 6b and S5e). Another fluorescence signal was observed in the base of the tentacles in one colony but was far less aggregated (Fig. S5a, b).

Transmission electron microscopy confirmed that bacteria resembling Mollicutes[61–64] were present in the mesoglea around the axis in five of six colonies (Fig. 5e–g) and in the mesoglea of polyps in four of six colonies (Fig. 6d, e). The mollicute cells lacked cell walls, were about 0.25 microns wide, and were pleomorphic with irregular shapes that were often elongated and possibly filamentous. These mollicutes were found in the acellular matrix of collagen fibers between tissue layers that form the mesoglea and were thus extracellular (Fig. 5g).

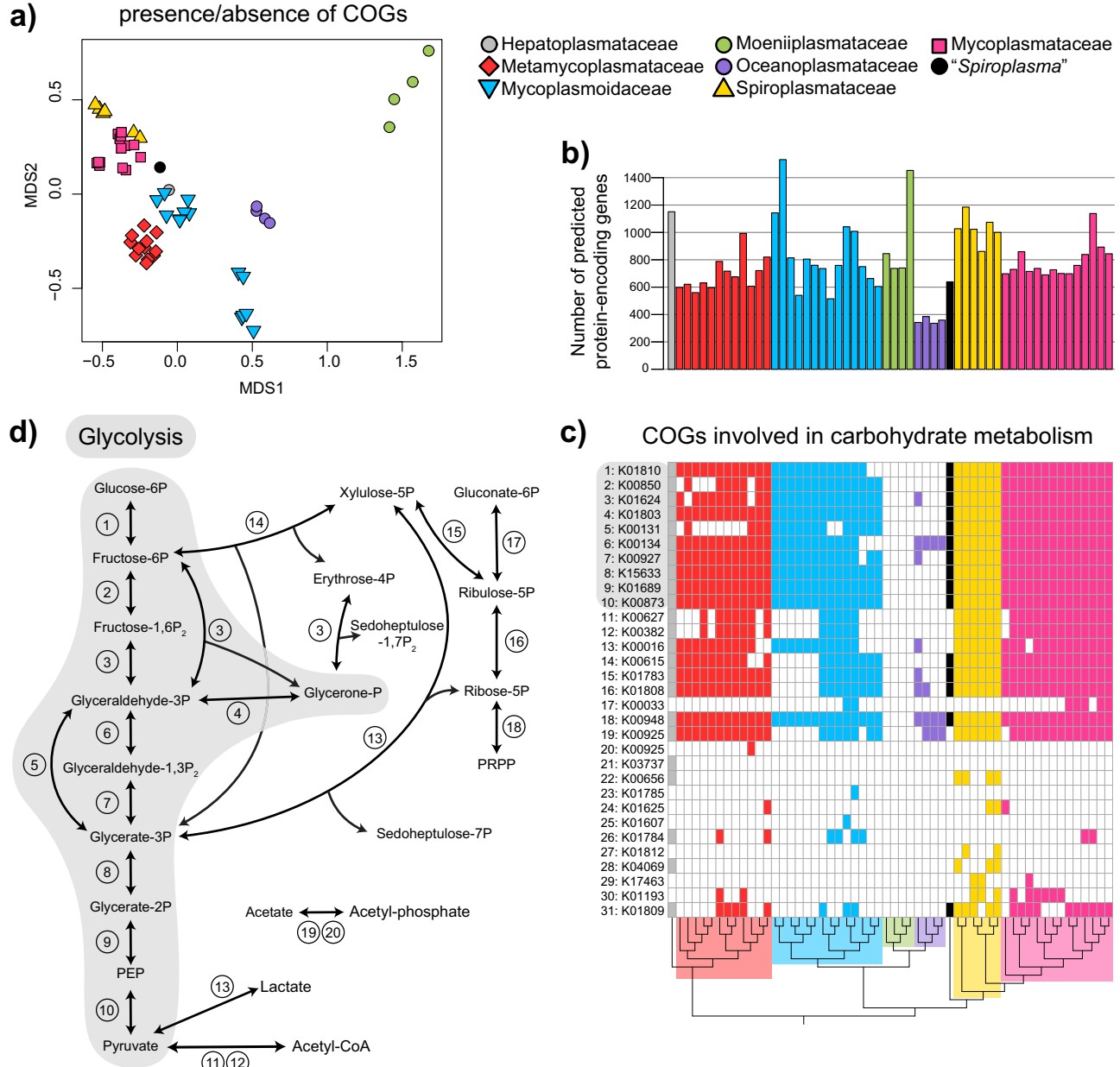

**Fig. 4 | Comparative Genomics of related mollicutes. a** Non-metric multi-dimensional scaling (NMDS) plot of clusters of orthologous genes (COG) from Mollicutes genomes using Jaccard distances. Only COGs present in more than two genomes were included. **b** The number of predicted protein-encoding genes for each genome. Oceanoplasmataceae had the fewest predicted protein-encoding genes. **c** The presence of COGs involved in carbohydrate metabolism for each genome. COGs are labeled using their KEGG orthology identifier annotations. **d** Metabolic map of glycolysis and other reactions associated with the COGs in (**c**). The map was drawn using reactions listed on the KEGG database for each KO identifier.

## Taxonomic descriptions

***Oceanoplasma* gen. nov.** (O.ce.a.no.plas'ma, L. masc. n. *oceanus*, the ocean, members of this genus are associated with marine invertebrates; Gr. neut. n. *plasma*, molded, lacking a cell wall, conventional suffix for mollicutes; N.L. neut. n. *Oceanoplasma*, wall-less bacteria of the ocean). See description for type taxon: *Ca. Oceanoplasma callogorgiae*.

***Oceanoplasma callogorgiae* sp. nov.** (ca.llo.gor'gi.ae, N.L. fem. n. *Callogorgia*, the genus of corals which harbor this bacterium; L. gen. singular neut. suff. *-ae*, of [*Callogorgia*]; N.L. gen. singular neut. n. *callogorgiae*, associated with corals in the genus *Callogorgia*). Non-fermentative mollicute associate of deep-sea octocoral, *Callogorgia delta*. Type material is NCBI Assembly: CP125803.

***Thalassoplasma* gen. nov.** (Tha.las.so.plas'ma, Gr. fem. n. *thalassa*, the sea, members of this genus are associated with marine invertebrates; Gr. neut. n. *plasma*, molded, lacking a cell wall, traditional suffix for mollicutes; N.L. neut. n. *Thalassoplasma*, wall-less bacteria of the sea). See description for type taxon: *Ca. Thalassoplasma callogorgiae*.

***Thalassoplasma callogorgiae* sp. nov.** Non-fermentative mollicute associate of deep-sea octocoral, *Callogorgia delta*. Type material is NCBI assembly JARVCM000000000.

***Oceanoplasmataceae* fam. nov.** (O.ce.a.no.plas.ma.ta.ce'ae. N.L. neut. n. *Oceanoplasma*, type genus of the family; *-aceae* conventional suffix to denote a family). This family consists of non-fermentative mollicutes that are associated with marine invertebrates. Their

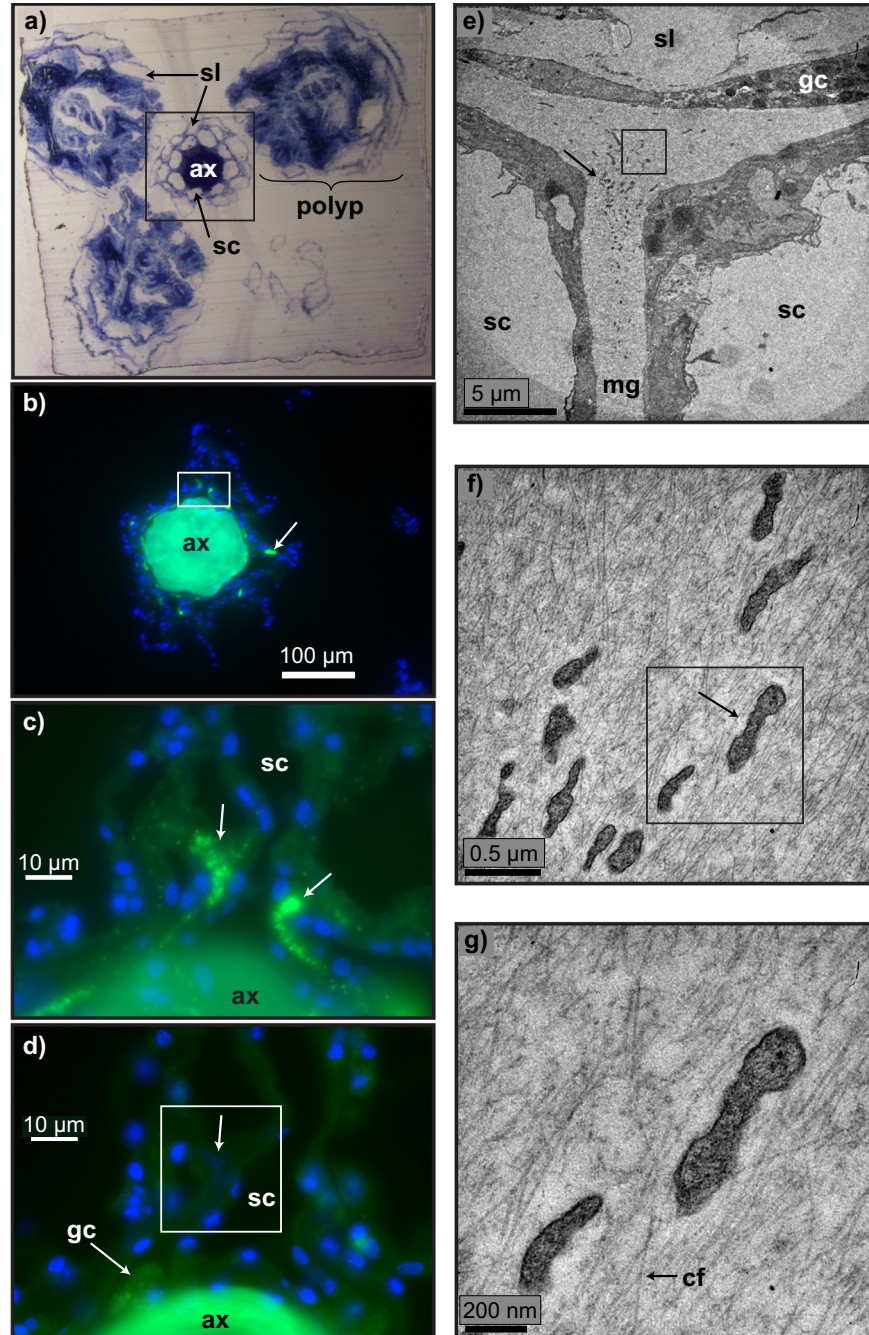

**Fig. 5 | Mollicutes in the mesoglea surrounding the axis in *Callogorgia delta*.**
**a** The orientation of polyps around the central axis shown in a thick section
(500 nm) embedded in Spurr's epoxy resin stained with toluidine blue and
methylene blue. **b**, **c** Bacterial signals in mesoglea near the axis in four of five
colonies from tissue sections hybridized using CARD-FISH with the general
eubacterial probe mix (EUB338 I-III, green) and stained with DAPI (blue). **d** Adjacent
section to (**c**) hybridized with negative control probe showing DAPI fluorescence
overlapping region where signal was detected. **e**–**g** Transmission electron micro-
graphs showing bacteria with characteristic mollicutes appearance seen in five of
six colonies. Unlabeled arrows indicate bacteria. Boxes indicate the exact region or
type of region next image shows. ax central axis, sl sclerite lacuna, sc stem canal, gc
granular cell, mg mesoglea, cf collagen fiber.

genomes are smaller than 500 kbp and lack the repertoire of genes
needed to conduct glycolysis but retain glyceraldehyde-3-phosphate
dehydrogenase. They can be distinguished from *Moeniiplasma* spp. by
the retention of ATP synthases and ribose-phosphate pyropho-
sphokinase. The type taxon is *Ca. Oceanoplasma callogorgiae*.

## Discussion
### Novel mollicutes in *Callogorgia delta* are symbionts
The data presented here suggests that *Ca. Oceanoplasma callogorgiae*
and *Ca. Thalassoplasma callogorgiae* are symbionts of *Callogorgia*

*delta sensu* Goff[65]. First, they were detected in *C. delta* colonies from all
sites and sampling years. Further, their genomes were very reduced
and demonstrated a reliance on compounds that are likely obtained
from *C. delta*. Finally, *Ca. Oceanoplasma callogorgiae* likely resides
within the coral. Abundant bacteria with mollicute-like morphology
were observed in the mesoglea of *C. delta* and *Ca. O. callogorgiae* was
the most abundant bacterium by far in both the amplicon dataset and
the metagenomes. Additional evidence to confirm the presence of *Ca.
O. callogorgiae* and *Ca. T. callogorgiae* in the coral mesoglea could
come from applying species-specific FISH probes.

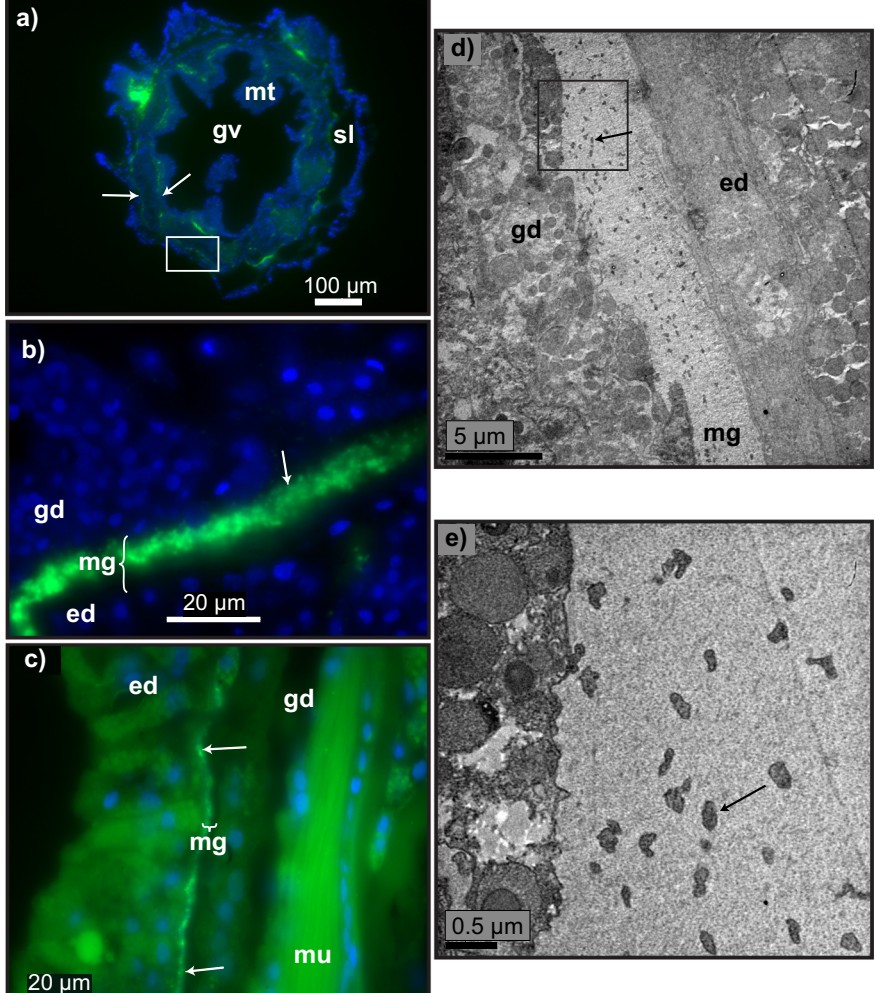

**Fig. 6 | Mollicutes in the mesoglea of polyps in *Callogorgia delta*. a** Bacterial signals in mesoglea of polyps seen in four of five colonies from tissue sections hybridized using CARD-FISH with the general eubacterial probe mix (EUB338 I-III, green) and stained with DAPI (blue). **a** Overview of an entire polyp in a transverse section, higher magnification (**b**), and a longitudinal section (**c**). **d**, **e** Transmission electron micrographs showing mollicutes-like bacteria in the mesoglea of polyps seen in four of six colonies. Unlabeled arrows indicate bacteria. Boxes indicate the exact region or type of region next image shows. gv gastrovascular cavity, mt mesentery, sl sclerite lacuna, gd gastrodermis, ed epidermis, mg mesoglea, mu muscle.

## Detection in sediment samples and transmission

The two novel mollicute species may associate specifically with corals in the genus *Callogorgia* since both were absent in the surrounding water and appear to be rare in sediment, as indicated by their low relative abundance. However, it is possible that Oceanoplasmataceae have a relatively high absolute abundance in the sediment but only comprise a low relative abundance because the total abundance of the entire microbial community may be high. Another possibility is that DNA from *Callogorgia* and its symbionts can be found in the sediment as environmental DNA originating from sources such as the fecal matter of grazers. Interestingly, eDNA from fish species is orders of magnitude more abundant in the sediment compared to the surrounding water and degrades slower[66,67]. Alternatively, their detection in sediment libraries may be due to low levels of contamination from *Callogorgia* samples such as through tag jumping or lane jumping. Up to five Molli-1 sequences were detected in three out of four sequencing blanks whereas it averaged 19 reads in sediment samples and over 7000 in *Callogorgia* libraries. See Supplementary Information for further details. Further experimental work is needed to confirm the prevalence of Oceanoplasmataceae in the sediment.

It is not clear how these symbionts are transmitted between coral colonies. Their streamlined genomes suggest they do not have a free-living stage. Therefore, it is possible that they are transmitted vertically through coral larvae. In addition, grazing fauna such as snails may serve as vectors transmitting symbionts between coral colonies. In shallow-water corals, corallivores including grazing gastropods are associated with the spread of bacterial and fungal pathogens[68]. Further, other mollicutes infect sessile hosts and are transmitted by mobile vectors, such as phytoplasmas, that infect plants and are spread by insects[69].

## Metabolic processes and other fundamental activities of the symbionts

Both *Ca. Oceanoplasma callogorgiae* and *Ca. Thalassoplasma callogorgiae* lacked genes encoding complete fermentative pathways and depended on arginine to generate ATP. They likely receive this arginine from their coral hosts as well as essential compounds such as amino acids, riboflavin, and biotin. Conversely, they export ornithine and possibly short peptides to their coral host.

*Ca. Oceanoplasma callogorgiae* possessed extensive mechanisms to defend against viruses or other forms of foreign DNA. Its genome contained an extensive CRISPR array and some of the most abundant transcripts in both of our libraries were restriction-modification systems and endonucleases. Further, it shared additional genomic

features with *Ca. Thalassoplasma callogorgiae* including *comEC* to import foreign DNA and enrichment of thymidine kinase genes that may salvage thymidine from DNA degradation. The presence of these antiviral characteristics in such highly reduced bacterial genomes suggests that they play a central role in the lifestyles of these bacteria.

Other, not yet well-understood processes likely underly the basic functioning of these novel mollicutes since some of the most highly transcribed genes in this study had unknown functions. This includes U1, which is likely membrane-bound and potentially degrades cyclic di-GMP. Cyclic di-GMP is a second messenger involved in morphogenesis, motility, and virulence[70]. U1 may respond to external or host-derived stimuli as part of signaling pathways that could be linked to host innate immunity and symbiont recognition. However, the metabolic insights gleaned from the most abundant transcripts in this study are limited by a low sample size ($n = 2$) which precludes standard statistical analyses and may not be reflective of general expression levels across populations.

While *Callogorgia delta* is found near cold seeps, neither *Ca. Oceanoplasma callogorgiae* nor *Ca. Thalassoplasma callogorgia* appear to have an association with seeps. Their genomes did not possess any genes or pathways that could utilize compounds from seeps for energy or nutrition, such as reduced sulfur species or hydrocarbons including light alkanes. Further, both mollicutes were also detected in *C. americana*, which is not found near cold seeps.

### Residence in the mesoglea

*Ca. O. callogorgiae* likely resides in the mesoglea of *C. delta*. The mesoglea of cnidarians is composed of a largely acellular, mucoid matrix of collagen containing some cells such as amoebocytes[71]. The mesoglea provides rigidity upon which the muscles of the polyp can act but also facilitates the transport of nutrients[72,73]. Some compounds diffuse through the mesoglea, like glucose and fatty acids[72,73], while amoebocytes can shuttle compounds such as amino acids[72]. *Ca. O. callogorgiae* probably acquires compounds including arginine, other amino acids, and the precursors of coenzymes such as riboflavin from the mesoglea because it lacks the genes necessary to synthesize them.

The mesoglea in cnidarians is also a site of immune defense[74–76]. Bacteria that reside in the mesoglea must evade the coral's immune defenses, such as phagocytic amoebocytes[74]. A few bacteria have been localized within the mesoglea of cnidarians, including *Pseudomonas* and spirochetes in *Hydra* spp.[77,78] and pathogenic cyanobacteria in the coral *Orbicella annularis* suffering from black band disease[79]. In contrast, the only mollicute previously localized in any coral was *Mycoplasma corallicola*, which was found on the surface of the tentacle ectoderm of *Lophelia pertusa*[41]. Unlike *M. corallicola*, *Ca. O. callogorgiae* likely resides within its host coral and therefore must somehow evade phagocytic amoebocytes in the mesoglea and potentially other immune defenses.

### Consequences for the coral host

It is still unclear if *Ca. Oceanoplasma callogorgiae* incurs a cost or provides any benefit to its coral host. These novel mollicutes may simply be commensals or parasites with a minimal impact on their host corals since they did not display any obvious pathology and nearly every colony appears to be infected. Conversely, the metabolism of arginine may provide the coral with an alternative pathway to process nitrogenous waste or could function to recycle nitrogen which would be useful for deep-sea corals because many rely on a nitrogen-poor diet of marine snow[80]. It is also possible that some of the unannotated genes are peptides that confer protection from pathogens or parasites like those that *Spiroplasma* spp. produce in insects[21]. Others have suggested that the symbionts of hadal sea cucumbers, identified herein as belonging to Oceanoplasmataceae, may provide protection to their hosts from animal viruses using their CRISPR-Cas and restriction-modification systems[24]. It is unclear if this

is a possibility for *Ca. O. callogorgiae* since it likely resides in the mesoglea where direct exposure to viruses may be low. Virus-like particles have been observed in the mesoglea of a shallow-water, stony coral[81], however they were rare and far less abundant than those found in the epidermis and gastrodermis. Exposure to viruses in the mesoglea may increase when predators graze on corals or when branches break. Indeed, gastropods have been observed grazing on *Callogorgia delta* colonies and they suffer higher rates of branch loss compared to other deep-sea corals in the genus *Paramuricea* which do not host Oceanoplasmataceae[82]. Further, upon injury, phagocytic amoebocytes migrate from the mesoglea to the wound where they provide defense against invading pathogens[83]. If *Ca. O. callogorgiae* has a role in immune defense, it may also reside in the mesoglea and migrate when tissue is wounded.

### The novel family Oceanoplasmataceae

We propose the family Oceanoplasmataceae, which includes *Ca. Oceanoplasma callogorgiae*, *Ca. Thalassoplasma callogorgiae*, and other associates of marine invertebrates. This is supported by the fact that it forms a highly divergent and well-supported monophyletic clade, it is ecologically distinct from other Mollicutes since all members associate with marine invertebrates, and it shares specific genomic features such as lacking the genes involved in glycolysis while retaining genes encoding ribose-phosphate pyrophosphokinase, glyceraldehyde-3-phosphate dehydrogenase, and ATP synthase.

The hosts of Oceanoplasmataceae are phylogenetically diverse comprising five animal phyla, exhibit diverse lifestyles from being photosynthetic to wood-borers, and originate from a wide range of habitats from the epipelagic zone to hadal trenches[24–28]. In addition to this, the large phylogenetic distances between members of the Oceanoplasmataceae suggests that a substantial amount of diversity remains undiscovered. Interestingly, *Ca. Moeniiplasma* spp. appear to be the closest relatives of Oceanoplasmataceae. They clustered together in the phylogenetic analyses and shared some genomic similarities, such as a lack of glycolysis.

Compared to other mollicutes, Oceanoplasmataceae exhibit extensive viral defense systems. All were enriched in thymidine kinase genes and the two genomes with unfragmented CRISPR arrays had more spacers than other Mollicutes. Only the associate of a hadal snailfish, *Mycoplasma liparidae*, had more spacers[53]. This may reflect a general evolutionary trend among animal-microbial symbioses in marine environments where exposure to viruses is high. Similarly, the symbionts of marine sponges are enriched in CRISPR-Cas systems compared to surrounding seawater[84,85].

Oceanoplasmataceae possess the most reduced genomes among all mollicutes. They possess the smallest genomes, which contain the fewest protein-encoding genes and lack any complete fermentative pathway. Extensive genome reduction dominates the evolutionary history of Mollicutes. Mollicute genomes are used to identify the minimum set of genes essential for cellular life[23] and are utilized as precursors in efforts to design minimal genomes[86,87]. These synthetic genomes retain glycolysis. However, the lack of glycolysis in Oceanoplasmataceae and *Moeniiplasma* spp. suggests that these synthetic genomes could be reduced further. Our comparative genomics analysis also revealed that *Moeniiplasma* spp. lack ATP synthase, suggesting alternative gene sets may permit further reduction.

Strangely, no mollicutes except *Acholeplasma laidlawii* are known to use cyclic di-GMP and all mollicutes instead use cyclic di-AMP[88]. The presence of an EAL domain in gene U1 of *Ca. O. callogorgiae* implies that the use of cyclic di-GMP was not lost in all other mollicutes or was regained through horizontal gene transfer in some lineages.

Here we describe two novel members of the Mollicutes that associate with the deep-sea coral, *Callogorgia delta*, and propose the names *Ca. Oceanoplasma callogorgiae* and *Ca. Thalassoplasma callogorgiae*. We characterize their association with *C. delta*, generate

genomic resources, and produce phylogenetic and metabolic inferences advancing our understanding of coral-associated mollicutes and symbiosis in the deep sea. Further, we propose a new family, Oceanoplasmataceae, which has not yet been recognized and whose diversity remains largely uncharacterized. We show that its genome reduction exceeds what was known among Mollicutes, informing their evolutionary history, the consequences of symbiosis, and the concept of minimal bacteria.

## Methods

### Collections

Collections are a subset of those described in ref. 5. One hundred and eight *Callogorgia delta* colonies were collected from six sites spanning over 350 km in the northern Gulf of Mexico in 2015, 2016, and 2017 (Fig. 1). The sites are named after the Bureau of Ocean Energy Management's designations for lease blocks in which corals were sampled and include Mississippi Canyon (MC) 751 (*n* = 19 colonies, 435-443 m, 28.193 N-89.800 W), Viosca Knoll (VK) 826 (*n* = 3 colonies, 547-559 m, 29.159 N-88.010 W), Green Canyon (GC) 234 (*n* = 37 colonies, 505-536 m, 27.746 N-91.122 W), MC885 (*n* = 25 colonies, 624-642 m, 28.064 N-89.718 W), GC249 (*n* = 14 colonies, 789-795 m, 27.724 N-90.514 W), and GC290 (*n* = 10 colonies, 852 m, 27.689 N-90.646 W). Within a site, corals spanned a range from within a meter apart to a maximum of 32-1514 m apart depending on site (Table S3 and Supplementary Data 1). To determine if novel mollicutes could also be detected in related coral species, we additionally processed four *Callogorgia americana* colonies that were collected in 2010 from Garden Banks (GB) 299 (*n* = 2 colonies, 358-359 m, 27.689 N-92.218 W) and VK862 (*n* = 2 colonies, 352-357 m, 29.109 N-88.387 W).

We obtained Letters of Acknowledgment from the National Marine Fisheries Service, Southeast Regional Office (263 13th Avenue South, St. Petersburg, Florida 33701-5505) and notified the Bureau of Ocean Mineral Management (Notice NG15-004) of scientific research prior to conducting sampling. The letters of Acknowledgment (LOA) recognized the activities as scientific research in accordance with the definitions and guidance at 50 CFR 600.10. As such, the proposed activities are not subject to fishing regulations at 50 CFR 622 developed in accordance with the Magnuson-Stevens Fishery Conservation and Management Act.

Colonies were sampled using specially designed coral cutters consisting of blades mounted on the manipulator arm of remotely operated vehicles (ROVs). Around 30 cm of the distal end of branches were removed, placed in separate temperature-insulated containers until recovery of the ROV. These consisted of coral quivers and bioboxes composed of high-density polyvinyl chloride (HD-PVC, Fig. S3). The coral quivers were capped cylinders (15 cm internal diameter by 25-40 cm tall) that were sealed with rubber stoppers (Fig. S3). The bioboxes were sealed with a rubber gasket and contained internal dividers. Biobox dimensions varied between 30.5-65 cm long, 30.5-50 cm wide, and 25-30.5 cm tall and were composed of 1.9 cm thick HD-PVC. Upon recovery, corals were maintained at 4 °C for up to 4 h until subsamples were flash-frozen in liquid nitrogen and fixed for microscopy (see below for details). Colonies collected in 2015 were preserved in 95% ethanol.

Sediment samples were taken in close proximity to many of the *Callogorgia delta* collections using push cores with a diameter of 6.3 cm. Upon recovery of the ROV, 1 mL of sediment from the top 1 cm of each sediment core was frozen in liquid nitrogen for later DNA extraction for microbiome analysis. Water was sampled in 2015 using a Large Volume Water Transfer System (McLane Laboratories Inc., Falmouth, MA) which filtered approximately 400 L of water through a 0.22 micron porosity filter with a diameter of 142 mm. Each filter was subsampled by preserving one quarter in ethanol and freezing another quarter in liquid nitrogen. In 2016 and 2017, 2.5 L Niskin bottles mounted on the ROV captured water above corals. Upon recovery of

the ROV, this water was filtered through 0.22 micron filters which were frozen in liquid nitrogen.

### 16S rRNA metabarcoding

The microbiomes of *Callogorgia delta, Callogorgia americana*, water, and sediment were analyzed using 16S metabarcoding. DNA was extracted from *C. delta* tissue and sediment samples using DNeasy PowerSoil kits (Qiagen, Hilden, Germany) following manufacturer protocols using ~1 cm of coral branches and about 0.25 g of sediment. DNeasy PowerSoil kits were also used for water samples collected in 2015 using 1 cm² of filter.

Other samples involved different processing methods. DNA was extracted from *C. americana* samples using DNeasy allprep kits with β-mercaptoethanol added to the RLT+ lysis buffer following the manufacturer's instructions. Subsamples from one colony were extracted using both kits. Additionally, subsamples from five *C. delta* colonies were processed using both kits. For all water samples from 2015, replicate extractions were performed on quarters of the filter preserved in both ethanol and frozen in liquid nitrogen. DNA was extracted from all other water samples using Qiagen DNeasy Power-Water kits. See Table S1 in the Supplementary Information for all details regarding processing methods for all samples. Some colonies were sampled multiple times across years and multiple times across the colony. Colonies were identified each year by location and unique branching pattern using images and geographic coordinates from previous collections. ASVs were considered detected in a colony if they were detected in at least one replicate of the colony. Only the first replicate from frozen tissue was used to report relative abundances or in Fig. 1.

The V1 and V2 regions of the 16S rRNA gene were amplified using universal bacterial primers 27F and 355R[89] with CS1 and CS2 adapters (Illumina, San Diego, CA). See Table S4 for primer sequences. PCR was conducted with the following reaction composition: 0.1 U/μL Gotaq (Promega, Madison, WI), 1× Gotaq buffer, 0.25 mM of each dNTP, 2.5 mM MgCl$_2$, and 0.25 μM of each primer, and the following thermocycler conditions: 95 °C for 5 min; 30 cycles of 95 °C for 30 s, 51 °C for 1 min, and 72 °C for 1 min; and finally 72 °C for 7 min. Libraries were prepared by the University of Illinois Chicago DNA services facility and sequenced on two separate runs on an Illumina MISEQ platform[90]. Raw sequence data are available on the NCBI database under Bio-Project PRJNA565265 and BioSample IDs SAMN12824733-4765, SAMN12824767-4875.

Amplicon sequence data were analyzed using QIIME 2 (ver2017.11)[91] following the Moving Pictures Tutorial with default parameters unless otherwise noted. Reads were joined using vsearch[92] and quality filtered using q-score-joined. Deblur[93] was used to detect chimeras, construct sub-operational taxonomic units (referred to as ASVs), and trim assembled reads to 300 bp. ASVs were classified using the SILVA 132 SSUref nr99 by first extracting the V1-V2 region of these reference sequences and truncating to 400 bp with extract-reads, creating a naive Bayes classifier using fit-classifier, then classifying ASVs using scikit-learn[94].

### Metagenomes and metatranscriptomes

Metagenomes and metatranscriptomes of *C. delta* were sequenced to assemble genomes of novel mollicutes which were used to assess their phylogenetic positions and metabolic potential. DNA was extracted from eight *Callogorgia delta* colonies as described in ref. 14 (collected in 2016, four colonies from MC751, two from MC885, two from GC234, Table S2). In brief, DNA was extracted using DNA/RNA allprep kits (Qiagen, Hilden, Germany) with β-mercaptoethanol added to the RLT+ lysis buffer following the manufacturer's instructions. These DNA extracts were sequenced on an Illumina HISEQ2500 platform using 150 bp paired-end reads. The two libraries with the highest coverage of mollicutes were sequenced a second time (Table S2). Raw sequence

data are available on the NCBI database under BioProject ID PRJNA574146 and BioSample IDs SAMN12856800–SAMN12856807.

In addition to these, DNA was extracted from an additional 18 samples corresponding to 16 additional colonies (Table S2). These were submitted to the Pennsylvania State University Genomics Core facility for library preparation and sequenced on an Illumina HISEQ platform with 100 bp paired-end reads.

RNA was extracted from two *Callogorgia delta* colonies (Table S2) using an RNeasy extraction kit (Qiagen, Hilden, Germany). The RNA extracts were enriched for bacteria by depleting coral host rRNA using a Ribo-Zero Gold Yeast kit (Illumina, San Diego, CA, USA) following the manufacturer's protocols. The RNA extracts were submitted to the Pennsylvania State University Genomics Core facility for library preparation and sequenced on a MISEQ platform with 150 bp single-end reads. Raw sequence data are available on the NCBI database under BioProject ID PRJNA565265 and SRA accession numbers SRR10174410 and SRR10174411.

All DNA sequence libraries were screened for bacterial SSU rRNA using phyloFlash ver3.3[95]. The two libraries with the highest coverage of mollicute 16S rRNA were sequenced a second time more deeply. Of these, the library with higher coverage was chosen for further processing. Reads were trimmed to a quality score of 2 from both ends using BBduk (BBtools ver37.52[96]) and a k-mer frequency analysis was performed using BBnorm. The library was k-filtered to a depth of 1000 or greater using BBnorm and downsized to 10% using reformat with a single pass. The library was then assembled using SPAdes ver3.11.0[97] with a k-mer progression of 21,33,55,77,99,127. Bandage ver0.8.1[98] was used to identify a single circular scaffold of contigs containing the dominant mollicute 16S rRNA gene. The overlaps on the end were manually trimmed off and the scaffold was recentered to the origin of replication predicted by Ori-Finder[99]. This constituted the draft genome for the dominant mollicute and was annotated using classic RAST ver2.0[100,101].

In order to generate a genome for the less abundant mollicute, the two libraries with the highest coverage of its 16S rRNA gene were coassembled using megahit ver1.1.2[102]. All libraries were mapped to this coassembly using BBmap with a kfilter of 22, subfilter of 15, and a maxindel of 80. Alignments were converted from SAM to BAM formats and sorted using samtools[103] and used for binning using MetaBat 2 ver2.12.1[59,104] with default parameters. All scaffolds with contigs belonging to bin 1 constituted a draft genome that was annotated using RAST and used for phylogenomic analysis.

The presence of CRISPRs and associated genes along with the number of spacers were determined using RAST annotations and CRISPRfinder ver4.2.30[105,106]. OAT: OrthoANI Tool ver0.93.1[107] was used to estimate average nucleotide identity between genomes. CRISPR spacer sequence from *Ca. Oceanoplasma callogorgiae* and *Ca. Thalassoplasma callogorgiae* were queried against the CRISPRCasdb[54] to identify potential matches. To determine which genes were transcribed and to quantify their transcription levels, the two RNA sequence libraries were pseudoaligned to the gene annotations of both genomes with kallisto ver0.44.0[108] using 100 bootstrap replicates, an estimated fragment length of 180 bp with a standard deviation of 20 bp, and an index with a k-mer size of 31 bp. The transcription levels of all genes were ranked to identify the genes with the highest transcription levels under the conditions present during sampling by sorting by mean transcripts per million (tpm). Among genes with non-zero mean transcription levels, the top eight most abundant transcripts were outliers after log-transformation (greater than the third quartile plus 1.5 times the interquartile range). The top five most abundant transcripts were outliers in each library individually. Coding regions among the ten most highly transcribed but lacking functional annotations from RAST or DRAM were further investigated using blastp and HHBlits[109] to inform potential function.

## Phylogenetic analyses

Phylogenetic trees were constructed to infer the phylogenetic positions of the mollicutes that were detected in *Callogorgia delta*. First a tree of 16S sequences was generated to place these novel mollicutes within the wide diversity of mollicutes with available 16S sequences (accessions in Supplementary Data 1). Full-length 16S rRNA genes were assembled from the metagenomes using phyloFlash ver3.3[95]. These full-length sequences were aligned with publicly available Mollicutes sequences that were over 1000 bp using MUSCLE ver3.8.425[110]. A maximum likelihood tree was constructed using the IQ-TREE web server ver1.6.10[111,112]. Within IQ-TREE, ModelFinder[113] was used to choose a general time reversible model with unequal rates and empirical base frequencies as well as a FreeRate model[114,115] for rate heterogeneity across sites with 3 categories (GTR + F + R3) based on the highest Bayesian information criterion (BIC) score. UFBoot2[116] was used to obtain ultrafast bootstrap support values (UFBoot) using 1000 replicates.

To obtain a more confident phylogenetic placement of the coral-associated mollicutes, a phylogenomic tree was constructed including genomes from the two mollicutes in *C. delta* and 66 mollicute genomes accessed through GenBank and the European Nucleotide Archive (ENA) (Supplementary Data 1). One more mollicute genome was assembled from a dataset available on ENA because an assembled genome was not available. This genome corresponded to the novel bacterium that associates with the jellyfish, *Cotylorhiza tuberculata*, which was tentatively identified as *Mycoplasma*[25]. Among the four libraries, M3 was chosen for assembly since it had the highest coverage of *Mycoplasma* based on the coverages of 16S rRNA assembled using phyloFlash[95]. This library was then assembled using megahit ver1.1.2[102] with a kmax of 241. All four libraries were mapped to this assembly using bbMap with a kfilter of 22 bp, subfilter of 15 bp, and maxindel of 80 bp. The alignments were subsequently used to bin the assembly using MetaBat 2[59,104] using default parameters. Bin one corresponded to the purported *Mycoplasma* genome. This bin and all associated scaffolds were used to represent this genome in phylogenomic analyses and was also annotated using RAST to compare to other genomes. A concatenated amino acid sequence alignment of 43 genes from all 69 genomes was generated with CheckM ver1.1.3[117]. A maximum likelihood phylogenomic tree was then constructed using IQ-TREE ver1.6.12[111,116]. Both phylogenetic trees were annotated using the recently revised phylogeny of the class Mollicutes[118–120].

Additional hosts of the family Oceanoplasmataceae were identified in publicly available 16S rRNA amplicon datasets using the integrated microbial NGS platform (IMNGS)[49]. Seven full-length 16S rRNA gene sequences representing the known breadth of Oceanoplasmataceae were queried against the IMNGS database using the lowest identity threshold (90%). These sequences represented *Ca. Oceanoplasma*, *Ca. Thalassoplasma*, and the symbionts from the five other hosts mentioned previously. Samples with related sequences that totaled more than 5% were considered potential hosts.

## Comparative genomics analyses

All mollicute genomes included in phylogenetic analyses that use translation table 4 (all except Acholeplasma-Anaeroplasma-Phytoplasma group) were annotated using DRAM[121]. Clusters of orthologous genes (COGs) were identified among all genomes using orthoMCL[122] on KBase with the BuildPangenome with OrthoMCL app (version 2.0) using amino acid translations from all protein-encoding genes (pegs) and default parameters. To visualize differences in genomic composition, non-metric multidimensional scaling was performed using Jaccard distances based on the presence/absence of COGs. Only COGs that were present in more than two genomes were included.

## Microscopy

CARD-FISH was employed to localize bacteria within *Callogorgia delta*. Colonies collected in 2016 were preserved for microscopy. During subsampling, 0.5 cm long segments from tips of branches were fixed in 2 mL of 4% formaldehyde with 1× phosphate-buffered saline (PBS) at 4 °C overnight (4–8 h). Samples were washed 3 times and then stored in equal parts ethanol and 1X PBS for up to 21 months. Samples were decalcified in 0.45 M EDTA in 1× PBS for 7 days, refreshing the buffer after 4 days, then washed in 1× PBS. Samples were postfixed in 1× PBS with 1% paraformaldehyde for 1 h then washed 3 times in 1× PBS and stored in 50:50 1× PBS:ethanol overnight. The next day, the samples were embedded in TissueTek (Sakura Finetek, Maumee, OH) and left at 4 °C overnight. The next day, the TissueTek was replaced and the samples were frozen. Samples were sectioned to a thickness of 4 μm using a Leica (Wetzlar, Germany) cryostat CM3050 S. Adjacent sections were placed on polysine slides and stored at 4 °C before hybridization.

Hybridization began by removing TissueTek from tissue sections via soaking slides in milliQ water for 3 min. Autofluorescence was then diminished by soaking slides in ethanol with 0.1% Sudan black B for 10 min. Sections were washed three times in 1× PBS and left in 1× PBS for a final 24 min. Endogenous peroxidases were inactivated by soaking sections in 0.2 M HCl for 12 min, then 20 mM Tris-HCl for 20 min, lysozyme solution (0.01 g/mL) for 30 min at 37 °C, 20 mM Tris-HCl again for 20 min, and finally 3 min in milliQ water. Slides were then air-dried and sections were circled with a PAP pen. Two probes were used: a mix of EUB338 I-III which target eubacteria[123,124] and a negative control probe (non-EUB) consisting of the reverse complements (see Table S4 for sequences). Hybridization mixes were created by combining 1 μL of probe (50 ng/μL) with 299 μL hybridization buffer (30% formamide with 0.102 M NaCl, 0.02 M Tris-HCl, 10% Blocking Reagent, 0.01% sodium dodecyl sulfate, and 0.1 g/mL dextran sulfate). Each section was covered with ~10–20 μL of one of the hybridization mixes. Two adjacent sections were hybridized with the EUB338 I-III probe mix and non-EUB probes separately to produce a negative control of the same tissue region (Figs. S4 and S5a–d). Each slide was then placed in a humidity chamber consisting of a closed 50 mL tube with a kimwipe soaked in 2 mL of 30% formamide in milliQ water. Hybridization was accomplished by incubating slides at 46 °C for 2 h within their humidity chambers.

After hybridization, slides were washed for 15 min at 48 °C in washing buffer (0.102 M NaCl, 0.02 M Tris-HCl, 0.005 M EDTA, 0.01% SDS) followed by 15 min in 1× PBS at room temperature. A fresh amplification mix was created by first combining 299 μL 1× PBS and 1 μL H₂O₂, then adding 10 μL of that solution to 1 mL of amplification buffer (1× PBS, 0.01% Blocking Reagent, 2 M NaCl, 0.1 g/mL dextran sulfate) and 1 μL of labeled tyramide (1 mg/mL ALEXA 488-labeled tyramide in dimethylformamide with 20 mg/mL p-iodophenylboronic acid). Slides were again placed in 50 mL humidity chambers with a kimwipe soaked in 2 mL of milliQ water. Amplification was accomplished by incubating slides at 37 °C for 30 min in these humidity chambers.

After amplification, slides were washed by dipping in 1× PBS followed by 10 min in 1× PBS, and a final dip in milliQ water. Slides were air-dried and sections were stained with DAPI (1 μg/mL) for 10 min in the dark. Slides were washed a final time in milliQ water for 3 min. Once air-dried, slides were mounted using Vectashield (Vector Laboratories, Burlingame, CA) and viewed on an Axioscope epifluorescence microscope (Carl Zeiss AG, Oberkochen, Germany) with filters restricted to the emission spectra of ALEXA 488 and DAPI. Images were taken with a Zeiss AxioCam camera and visualized using AxioVision software. CARD-FISH signals were considered positive if they appeared consistently within a structure in the coral, were absent in the same structures in negative control sections, and overlapped DAPI signals (Figs. S4 and S5a–d).

During subsampling, 0.5 cm long segments from the tips of branches were fixed in 2.5% (vol/vol) glutaraldehyde, 9% (wt/vol) sucrose, and 1.5× PHEM buffer (90 mM PIPES, 37.5 mM HEPES, 15 mM EGTA, 3 mM MgCl₂) at 4 °C overnight (~8 h). Samples were then washed with 1.5× PHEM and 9% sucrose thrice and stored in the final wash buffer. Samples were decalcified 2–2.5 weeks later using 0.4 M EDTA, 1.5× PHEM, and 9% sucrose for 3 days then washed three times with 1.5× PHEM in 9% sucrose. Samples were then washed twice in 0.1 M Na cacodylate buffer and postfixed for 1 h in 1–2% osmium tetroxide in 0.1 M Na cacodylate buffer. Samples were then washed twice in cacodylate buffer and then with milliQ water. Samples were then stained en bloc with 2% aqueous uranyl acetate overnight then dehydrated with the following ethanol series: 25%, 50%, 70%, 85%, 90%, 95%, 100%. Next, samples were further dehydrated with three washes of 100% molecular grade ethanol and three acetone washes. Then samples were infiltrated with a 50:50 mix of acetone and Spurr's epoxy resin for 24 h, then 100% Spurr's epoxy resin for 24 h three times. Finally, samples were placed in molds, filled with fresh Spurr's epoxy resin then baked at 60 °C overnight to polymerize blocks.

Blocks from six *Callogorgia delta* colonies were sectioned on a Leica EM UC6 Ultramicrotome (Leica Microsystems, Wetzlar, Germany) using an Ultra 45° diamond knife (DiATOME, Nidau, Switzerland) to a thickness of 70 nm. Sections were retrieved on 200 mesh copper grids with formvar and carbon coating and viewed on a JEM-2100 Electron Microscope (JEOL, Tokyo, Japan) with an accelerating voltage of 120 kV. To provide an overview of tissue morphology, thick sections (500 nm) were stained with epoxy tissue stain (toluidine blue and basic fuchsin) and viewed with a Nikon Eclipse E1000 microscope (Nikon Corporation, Tokyo, Japan).

### Reporting summary

Further information on research design is available in the Nature Portfolio Reporting Summary linked to this article.

## Data availability

Raw sequence data generated or used in this study are available on the NCBI Sequence Read Archive under BioProjects PRJNA574146 (BioSample IDs SAMN12856800–SAMN12856807) and PRJNA565265 (SRA accession numbers SRR10174410 and SRR10174411). Assembled 16S rRNA sequences and genomes are available under accession numbers OR679038−62, CP125803, and JARVCM000000000. The raw microscopy images generated in this study are deposited in Figshare: FISH, https://doi.org/10.6084/m9.figshare.26800249; thick section images, https://doi.org/10.6084/10.6084/m9.figshare.26801371; and TEM, https://doi.org/10.6084/10.6084/m9.figshare.26800081. Data used to construct figures are provided in Supplementary Data 1. The ASV classifications and ASV count table generated in this study are deposited in Figshare: https://doi.org/10.6084/m9.figshare.26837077.

## Code availability

No custom scripts were developed for this manuscript that are central to the main findings. Code used in this manuscript followed manual guidelines with details described in "Methods". Scripts are available upon request from the corresponding author.

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

## Acknowledgements

The authors would like to thank Martina Meyer, Miriam Sadowski, Silke Wetzel, and Meghann Devlin-Durante for their help with microscopy and DNA extractions. The authors would also like to thank the crews of all research vessels and ROV teams involved in collecting samples. This research was made possible by a grant from the Gulf of Mexico Research Initiative (GOMRI) which was awarded to the Ecosystem Impacts of Oil and Gas Inputs to the Gulf (ECOGIG) consortium (I.B.B. and C.R.F.). Additional funding was provided from the Max Planck Society (N.D.) and the National Academies of Sciences, Engineering, and Medicine Gulf Research Program Early-Career Fellowship under award 2000013668 (S.H.). Sample collections were originally funded by GOMRI through ECOGIG (I.B.B. and C.R.F.), the National Oceanic and Atmospheric Administration (NOAA) RESTORE Science Program under award NA17NOS4510096 (S.H.) and Bureau of Ocean Energy Management (BOEM) contracts 1435-01-05-CT-39187 and M08PC20038 to TDI-Brooks. H.G.V. was partially funded by the Deutsche Forschungsgemeinschaft (DFG; Heisenberggrant GR 5028/1-1). We acknowledge support by the Open Access publication fund of the Alfred-Wegener-Institut Helmholtz-Zentrum für Polar- und Meeresforschung. The funders had no role in study design, data collection and analysis, decision to publish, or preparation of the manuscript.

## Author contributions

S.A.V. and I.B.B. conducted sample collection, sample processing, and FISH microscopy. S.A.V. conducted bioinformatics analyses and TEM microscopy. I.B.B. and C.R.F. acquired funding, led research cruises, and supervised work. S.H. contributed additional funding. N.D. contributed additional funding and supervised work. H.G.V. advised bioinformatics analyses. S.A.V. wrote the manuscript. All authors reviewed and edited the manuscript.

## Funding

## Competing interests

The authors declare no competing interests.
