## [Peer Review file · Nature Communications]

Discovery of deep-sea coral symbionts from a novel clade of marine bacteria with severely reduced genomes

Corresponding Author: Professor Iliana Baums

Version 0:

Reviewer comments:

Reviewer #1

(Remarks to the Author)

The manuscript "Discovery of deep-sea coral symbionts from a novel family of marine bacteria, Oceanoplasmataceae, with severely reduced genomes" erect a new and interesting novel family that includes symbionts dominating the microbiome associated with the mesoglea of the Octocoral *Callogorgia delta*. These symbionts are part of a group presenting the most highly reduced genome among mollicutes. Although I agree that the authors present some exciting data on this new group and perform an interesting survey to provide preliminary insight into the potential role of these symbionts, I also have some major concerns about the comparison between the microbiome of samples collected and obtained using different tools, DNA extraction kits and sometimes even lack a minimum number of replicates (i.e., at least $n=3$). I believe the narrative is confusing, combining samples and goals that seem to have been originally obtained/obtained for a different purpose - which may perhaps dilute the strengths of the manuscript. I, therefore, have some suggestions that may contribute to improving the manuscript.

The abstract is well-written but not very clear about what is new about the group of symbionts described in this work and other already known mollicutes. From the introduction and overall literature, I understand mollicutes have already been described associated with octocorals, and many other marine hosts, and that their reduced genome is already known. I would focus and clearly highlight the new insights provided by the presented research, considering what is in the literature and the knowledge on the distribution of *Mycoplasma* spp.

Line 26: Reference 1 is an old reference to represent what is currently known about the diversity of microorganisms associated with corals, as research on coral-associated microbiomes has highly developed over the last seven years (and even more if we consider the following statement, "Most of these microbes were identified in Scleractinian corals from the photic zone...". Please consider including some more recent references. Although I understand the focus is on the newly erected family, if authors decide to keep the ecological survey, I also think the introduction should include, even if summarized, more data on the state-of-the-art of what is already known about the microbiome associated with deep-sea octocorals. For that, some key recent papers are missing, for example:
Pratte, Z.A., Stewart, F.J. & Kellogg, C.A. Functional gene composition and metabolic potential of deep-sea coral-associated microbial communities. *Coral Reefs* 42, 1011–1023 (2023). <https://doi.org/10.1007/s00338-023-02409-0>

Line 37: commonly?

Lines 67-79 and methods: While I acknowledge the limitations associated with the collection of deep-sea samples, I also struggle to associate the survey presented with the conclusions on the prevalence and specificity/comparison with environmental samples. Although the data presented is an interesting preliminary survey on the detection of these microorganisms in deep-sea samples collected on different cruises and years, the same group and other groups may have environmental and biological data collected using the same tools, from the same sites at the same sampling time, and processed using the same protocols, especially considering how microbiomes can respond to environmental changes and are differentially recovered when using different DNA extraction kits. I would perhaps focus this manuscript on the genomic and localization analyses and use the obtained data to guide a more well-planned ecological search for these Mollicutes, and the evaluation of their specificity across biological and environmental samples collected and processed using the same methods. More specifically:

Lines 82-86: The detection of this Mollicutes in different samples and different years is an interesting data. I would avoid comparing relative abundances, distributions and specificities (lines 86-88) unless the sampling design of the samples at very specific locations and the processing of the samples can be clarified (and I would not compare it with samples processed using different sampling tools and DNA extraction protocols).

Lines 225-231: How and from how many samples was this expression evaluated? How can this be considered high (compared to what and using what statistical methods)? More details are required regarding methods, standards and goals of using the RNA data.

Lines 324-327: Please provide images and more details of the specially designed tools and containers.

Lines 320-323: I would not keep this comparison due to the very low number of replicates (i.e., n=3 and even n=2 in one of the locations) from sites that seem quite far from each other.

Lines 331-337: The samples should be ideally collected using the same tools and strategies. Why were water samples collected using different methods?

Lines 342-343: If the DNA was extracted using different kits (on top of the different sampling processes/tools) these results are not comparable.

Lines 363-366: From which colonies? Collected in the same year, same sites? Please add more details for this entire section.

Lines 370-372: Same as above, it is not clear from which colonies, collected where and when, these metagenomes were obtained and why. Although I understand the goal was to assemble the genome, this information still needs to be included.

Line 373 and lines 397-403: Although a low number of replicates was acceptable for RNA analysis a few years ago, I don't think reporting data without a minimum n=3 is still acceptable.

Lines 397-403: What criteria was used for a specific expression to be considered "high"?

Lines 416-443: The comparative genomic analyses of the obtained genomes should probably be the focus of the paper, as the ecological aspects and comparisons seem rather disconnected at times and also lack a more appropriate experimental design and standardized processing of the samples.

Lines 443-481: Where Mollicutes previously located in the tissue of other deep-sea octocorals? If not, this is another strength of the paper, which would present the comparative genomic analyses and location of mollicutes, including new representatives, in dee-sea octocorals. The resolution of the figures could be improved, but I am unsure if this is an artifact of the built pdf.

Overall, a detailed table or figure explaining the sample collection, a map indicating the sampling sites and distances, the number of colonies collected each year and the proximity between colonies collected in different years and how and when environmental samples were collected would be very useful.

Reviewer #2

(Remarks to the Author)

Here, the authors discover a novel family of Mollicutes, herein named Oceanoplasmataceae, that was frequently associated with deep-sea corals as detected via 16S rRNA gene surveys and metagenomics. They also used fluorescence in situ hybridization to show that the corals host a dense community of bacteria associated with their tissues.

This paper was extremely well-written, easy to read and understand, and the figures beautifully done. The methods are impeccable. The 16S rRNA gene and metagenomic data strongly suggests that these bacteria are abundantly associated with the tissue. The authors were able to construct complete assemblies for these extraordinarily streamlined genomes, allowing for hypotheses about how they interact with the coral hosts.

However, this paper is significantly weakened by the lack of FISH evidence specific to the new group found here – I think it's reasonable to hypothesize based on the abundant bacterial populations (as observed via FISH) and the 16S and metagenomics data that these are associated with the tissue, but specific FISH probes would dramatically strengthen the claim. They also do not present TEM, which also seems important to their argument (if it is possible to do for the mesoglea) – for example to show the presence of cells with the characteristic shapes of Mollicutes (pear shaped, spiral shaped). The streamlined genome suggests an obligately host-associated lifestyle, which can be achieved through vertical transmission or through individual-to-individual lateral transmission with no free-living environmental stage. What is the

model for transmission here given the host is sessile (meaning the potential for direct inter-individual transmission seems low)? And how is that reconciled with the presence of Oceanoplasmataceae in the sediment samples?

Specific comments:

Line 170: The sentence regarding 16S coverage and binning is not clear, please rephrase for clarity.

Line 209-210: This feels like too strong of a statement to make at this point, that these are specific to Callagorgia, based on a few specimens and the environmental samples present here. Have you searched publicly available 16S datasets for the presence of this group? This can be done reasonably simply with the tool IMNGS (<https://doi.org/10.1038/srep33721>).

Line 263: Do you mean protection from animal viruses? If so, can you use the CRISPR spacers to determine what types of viruses they were interacting with?

Reviewer #3

(Remarks to the Author)

Overall:

This study documents a novel family of Oceanoplasmataceae that are associated with deep sea octocoral species. Through 16S rRNA gene sequencing, metagenomic reconstruction and fluorescent in situ hybridization visualization, the novel taxa are described and putative functions assigned. Overall, the manuscript presents an interesting study and well conducted experimental approach with valid and interesting findings. Minor revisions are recommended.

The 16S rRNA specific primers used in this study are valid though not commonly used across microbiome studies. One query is the coverage of these primers for the bacterial taxa? Do the primers miss certain groups and hence is the dominance of the Oceanoplasmataceae artificially elevated through primers not targeting certain common octocoral lineages? This information could be added in the supplementary section potentially. The metagenomic approach nicely supports the dominance of the novel Mollicutes clade, though adding further confidence that the 16S profiles are representative of the true octocoral associated microbial community would be advantageous.

The 16S rRNA gene bacterial sequence data primarily focuses on the Mollicutes affiliated sequences. There is no presentation of other bacterial taxa identified from the 16S data, some reference to this other diversity profile of the octocorals would be informative. Are these other sequences affiliated with common marine invertebrate associates as well?

Are certain energy sources for the novel Oceanoplasmataceae family derived directly from hydrocarbon seeps, which supports their growth. From the genome reconstruction for example are there metabolic pathways that may indicate metabolism of reduced chemicals such as methane or hydrogen sulfide if the host is found near these seep areas (as stated in the manuscript)? Also though oxygen is low at deep sites is there some level of oxidative pathways supporting growth?

Overall the data presented is robust and compelling; the only place where some further information or discussion would be valuable is the localization work. Generic Eubacteria probes were used for this visualization and while it is likely this detects the novel Mycoplasma taxa, you can not be certain all the detected signals are from this novel Oceanoplasmataceae group. While 16S and metagenomic information supports that they are the dominant taxa, it is an assumption that is being made and should be qualified. In addition it would be nice to show high resolution histological images side by side with the FISH images. The resolution of the FISH images are not great and the histological images can help resolve the location of the bacteria in the octocoral tissues. It stated that the bacterial cells are within mesoglea tissue regions though this can not be established from the FISH images. Specific stains (i.e. H&E, Gram, Mallory) can resolve the bacterial localization relative to the host tissues structures. The FISH images then confirm bacteria and localization within the mesogleal tissues layers. Currently just based on the FISH images, there is poor resolution of bacterial cells in relation to host tissues structures.

Other Specific points:

- Line 14: provide example of what mechanisms are used (though abstract maybe word limited, if there is capacity additional information here can help the reader).
- Line 59: Provide a reference to support statement of coevolution of Mycoplasma taxa with their host; or detail more explicitly this evidence from the references in the previous sentence.
- Line 200-203: The image resolution is quite poor for figures 5 and 6, especially in relation to the host tissue structure. Providing histology paired images with the FISH images would help resolved localization. For example, the cells are stated to be in aggregates, though this is hard to visualize in the images. Are these like CAMAs observed previously in scleractinian corals?
- Line 209-211: However, sequences of these novel taxa were commonly retrieved also from the sediments (as reported in lines 98-100; albeit at low abundance). Hence this statement needs to be refined. Their low abundance retrieval from sediments is partly influenced by the higher diversity of sediment samples; hence not really supported in concluding they are specific to the corals as could be some free-living stage as well. This goes to the previous comment made - are there potential energy sources from seep-environments that also support these novel lineages?
- Line 216: Not clear what is meant by "genomic coverage of the symbiont was up to 50 times as high as that of the host". Is that obtained reads were 50x more than the host? That seems unlikely unless some form of enrichment performed? Statement is unclear.
- Note: line 222 is repeating this statement "likely receive this arginine from their coral hosts" that is also stated above (lines

211-213).

- Line 231: why if host associated (and in aggregates) do they need these CRISPR systems? Does this indicate a potential free-living life stage (despite reduced genomes)? Would high levels of viral challenge be expected if residing solely in host tissues. Acknowledge sponge symbionts are enriched in CRISPR-Cas systems (line 284) but this makes sense since large amounts of water are passing through sponges and encounter rates with viruses and other microbes would be high in a sponge, but not in octocoral tissues.
- Line 262: Though as stated previously (line 74); they reside near hydrocarbon seeps which can be the predominant source of nutrients?
- Line 327: were samples also preserved in fixative's for FISH work?
- Line 342: Can any more detail be provided on where on the colonies these branches were taken? i.e. Tips of branches?
- Line 348: What is the coverage of these primers? This region of the 16S is less used than others for metabarcoding studies, hence some idea of coverage would be useful to demonstrate that major taxa clades are adequately covered.
- Figure 1: The bathymetric map is not a clear way to present the sampling sites i.e. contours are not informative and would suggest a stylized drawn B&W image with the same information of spatial location, site name, depth and number of sample taken from each site would be a better representation. Note in figure legend – states a star signifies water sampling; though do not see the stars.
- Figure 1: Why not provide an image of Callogaorai america as well?
- Figure 1; highlights that the 16S libraries of some samples are not dominated by the Oceanoplasma taxa. Hence what are the profiles of these samples and more broadly what other sequences were retrieved across all samples. See earlier comment on documenting at least briefly the extent of the diversity of these octocorals samples. Accept not the focus of the study but the Oceanoplasma seem not to be universally dominant in all samples (though most).
- Figure 5 and 6 could be consolidated easily into one Figure, as there is redundancy in many of the images for the specific details that are informative. See earlier comment though that good histological images would resolve the host tissue structure and microbiome localization much more clearly. For example, good histological staining will identify the mesoglea unambiguously and refine the polyp structures such as mesenteries and tissue layers (endoderm and gastroderm).

Version 1:

Reviewer comments:

Reviewer #1

(Remarks to the Author)

Thank you for the opportunity to review the revised version of your manuscript. I appreciate the efforts made to address the initial concerns. However, despite these revisions, there are still significant areas in the manuscript that require improvement, particularly concerning the methodological rigor and depth of the data analysis, which are critical for meeting the publication standards of Nature Communications.

The use of varied methods and sample collections across different times and conditions, coupled with the insufficient number of replicates, especially for the RNA analysis, raises substantial concerns about the consistency and reliability of the data. While I understand that differently processed samples can be employed for detecting specific groups, the potential biases introduced by these methodological inconsistencies are not adequately acknowledged. The use of unreplicated RNA data to indicate "high expression" still needs clearer justification, as typically, such claims should be supported by statistical analysis against controls. If the intent was to indicate mere "activity" or presence of gene expression, this should be explicitly clarified to avoid misleading interpretations.

The manuscript lacks critical FISH evidence needed to confirm the symbiotic interactions of the bacteria within coral tissues. Such visualization is essential to support some of the most important and ambitious claims of the manuscript. Additionally, I agree with the other reviewers that the ecological interpretations drawn from 16S sequencing data need further support, considering the limitations of these short sequences, the variability in sample processing and potential underestimation of the presence of Oceanoplasmataceae in sediments and the brief claims on their transmission - and I am not sure the revised text has fully addressed these concerns.

The discussion regarding the prevalence and ecological roles of these mollicutes seem a bit overstated given the evidence provided. A more cautious approach in discussing these findings is advised, particularly in light of the methodological issues noted above. In conclusion, while the revisions have introduced some improvements and the foundational findings hold promise, the manuscript still does not fully meet the rigorous standards required for publication in Nature Communications due to ongoing issues with methodological consistency, data interpretation, and provided evidence. I recommend substantial revisions to enhance the clarity of the conclusions that can be supported by the data.

Reviewer #2

(Remarks to the Author)

The authors did a great job of addressing my comments and suggestions. I'm especially appreciative of the new TEM data, which I think adds to their evidence.

My remaining comments are minor:

Lines 136-138: Clarify what threshold you used for IMNGS? You have in Methods but might be good for readers to also see that here to give context.

Lines 185-188: This information is not clear as to why it's relevant. Why is finding spacers also present in other bacteria relevant here? Instead, wouldn't it be more interesting to see what phages these spacers target? Especially given later discussion regarding whether these bacteria could play a role in viral defense for their coral host.

Line 213: Do you mean highest coverage of 16S rRNA in amplicon analysis of same sample? Or do you mean from the metagenome? Please clarify.

Reviewer #3

(Remarks to the Author)

The authors have addressed all concerns raised through the review process and therefore have no additional comments. This is a nice study and well deserving of publication in Nature Communications

Version 2:

Reviewer comments:

Reviewer #1

(Remarks to the Author)

I appreciate the authors effort to improve the manuscript and be accurate regarding the conclusions. The new version is indeed much better and mostly reflects the obtained data. I still disagree the use of duplicated samples is acceptable, and I believe the authors also agree with me, considering their response "Further, the discussion contains a statement that explicitly warns that the transcript levels observed in these two libraries may not be reflective of expression patterns generally because of the low sample size.". Although I understand the goal and reasoning, the levels are still questionable as they can be random/biased, which is why a minimal n of 3 is required to ensure reproducibility. It would be probably better to exclude this part, in my opinion.

Thank you to all reviewers for the time and effort you have put into reviewing our manuscript. Addressing your comments has greatly improved this manuscript. All reviewer comments are presented below in blue text. They have been numbered and are preceded by “Comment x.y” where x is the reviewer and y is their yth comment. Author responses follow each comment in black text preceded by the label “Response x.y”. Text from the manuscript is presented in quotes and underlined. Differences compared to the first submission is highlighted. Fully highlighted text represent completely new sentences. Line numbers provided by the authors when “all markup” is selected in the manuscript with tracked changes.

REVIEWER COMMENTS

Reviewer 1:

Comment 1.1 The abstract is well-written but not very clear about what is new about the group of symbionts described in this work and other already known mollicutes. From the introduction and overall literature, I understand mollicutes have already been described associated with octocorals, and many other marine hosts, and that their reduced genome is already known. I would focus and clearly highlight the new insights provided by the presented research, considering what is in the literature and the knowledge on the distribution of *Mycoplasma* spp.

Response 1.1 One of the major findings was that Oceanoplasmataceae (along with *Moeniiplasma*) are the only mollicutes that lack all known fermentative capabilities including glycolysis. We changed the abstract to make this clearer.

Abstract Line 12-14: “Unlike other mollicutes, they lack all known fermentative capabilities, including glycolysis, and can only generate energy from arginine provided by the coral host.”

This is reiterated by the final line of the abstract.

Abstract Line 20-22: “Oceanoplasmataceae genomes are the most highly reduced among mollicutes, providing new insight into their reductive evolution and the roles of coral symbionts.”

Comment 1.2 Line 26: Reference 1 is an old reference to represent what is currently known about the diversity of microorganisms associated with corals, as research on coral-associated microbiomes has highly developed over the last seven years (and even more if we consider the following statement, “Most of these microbes were identified in Scleractinian corals from the photic zone...”). Please consider including some more recent references. Although I understand the focus is on the newly erected family, if authors decide to keep the ecological survey, I also think the introduction should include, even if summarized, more data on the state-of-the-art of what is already known about the microbiome associated with deep-sea octocorals. For that, some key recent papers are missing, for example:

Pratte, Z.A., Stewart, F.J. & Kellogg, C.A. Functional gene composition and metabolic potential of deep-sea coral-associated microbial communities. *Coral Reefs* 42, 1011–1023 (2023).

Response 1.2 We have updated the citation to the most recent review that was published earlier this year. We added a citation to an octocoral microbiome review to support our statements about octocorals.

1. Voolstra, C.R., Raina, J.B., Dörr, M., Cárdenas, A., Pogoreutz, C., Silveira, C.B., Mohamed, A.R., Bourne, D.G., Luo, H., Amin, S.A. and Peixoto, R.S., 2024. The coral microbiome in sickness, in health and in a changing world. *Nature Reviews Microbiology*, pp.1-16.
4. van de Water, J.A., Allemand, D. and Ferrier-Pagès, C., 2018. Host-microbe interactions octocoral holobionts-recent advances and perspectives. *Microbiome*, 6, pp.1-28.

Introduction Line 28-34: “Corals also associate with a diversity of microbes¹, including the well-studied algal symbionts of the family Symbiodiniaceae and other microbial taxa that perform a variety of roles, from providing nitrogen through fixation² to causing disease³. Most of these microbes were identified in scleractinian corals from the photic zone¹, while studies investigating the roles of microbes in octocorals⁴ and deep-sea corals are rarer because of the limitations that great depths impose on sampling and experimentation.”

We have added the citation to this study on deep-sea corals and mentioned its results that are relevant to the introduction.

Introduction Line 37-40: “However, one study comparing deep-sea coral microbiomes to those in shallow-water corals found differences in metabolic activities including increased anaerobic ammonia oxidation and increased chitin degradation¹³.”

Comment 1.3 Line 37: commonly?

Response 1.3 The next paragraph explains what was meant by “commonly” but we removed the word to avoid confusion.

Introduction Line 40-43: “In addition, deep-sea octocorals also host associates that are rare or absent in shallow-water and/or scleractinian corals. These associates include bacteria from the SUP05 cluster, whose role is linked to cold seeps in the deep sea¹⁴, and commonly members of the class Mollicutes.”

Comment 1.4 Lines 67-79 and methods: While I acknowledge the limitations associated with the collection of deep-sea samples, I also struggle to associate the survey presented with the conclusions on the prevalence and specificity/comparison with environmental samples. Although the data presented is an interesting preliminary survey on the detection of these microorganisms in deep-sea samples collected on different cruises and years, the same group and other groups may have environmental and biological data collected using the same tools, from the same sites at the same sampling time, and processed using the same protocols, especially considering how

microbiomes can respond to environmental changes and are differentially recovered when using different DNA extraction kits. I would perhaps focus this manuscript on the genomic and localization analyses and use the obtained data to guide a more well-planned ecological search for these Mollicutes, and the evaluation of their specificity across biological and environmental samples collected and processed using the same methods.

Response 1.4 We agree that there are limitations when sample sizes are low and different methods are used such as different preservation methods and extraction kits. This can prevent comparisons of relative abundance in microbiome data or comparisons of prevalence. However, that was not the aim of this survey and we made no such inappropriate comparisons. The point of the survey was to determine the extent to which these mollicutes could be detected in Callogorgia samples and their surroundings. That is why we included all samples relevant to Callogorgia delta. The vast majority were frozen and extracted with powersoil kits however we utilized all our samples which included some that were preserved in ethanol and some that were extracted with allprep kits. This dataset demonstrated that Oceanoplasmataceae could be detected in samples from every site, from every year, as well as in Callogorgia americana. Further, the variation in methods show that Oceanoplasma could be detected in samples both preserved in ethanol and frozen as well as from DNA extracted using two different kits (DNeasy powersoil and DNeasy allprep). Further, the variation in methods strengthened some of the findings. Specifically, Oceanoplasmataceae was not detected in any water sample despite applying two different collection methods: niskins and a McLane pump.

In order to address these concerns, we made several changes to clarify the details of all the samples. In the methods, we added extra details to communicate that the majority of samples were frozen tissue extracted using powersoil kits and wherever that deviated.

Methods Section 467-503: “Upon recovery, corals were maintained at 4 °C for up to 4 hours until subsamples were flash frozen in liquid nitrogen and fixed for microscopy (see below for details). Colonies collected in 2015 were preserved in 95% ethanol.

Sediment samples were taken in close proximity to many of the *Callogorgia delta* collections using push cores with a diameter of 6.3 cm. Upon recovery of the ROV, 1 mL of sediment from the top 1 cm of each sediment core was frozen in liquid nitrogen for later DNA extraction for microbiome analysis. Water was sampled in 2015 using a Large Volume Water Transfer System (McLane Laboratories Inc., Falmouth, MA) which filtered approximately 400 L of water through a 0.22 micron porosity filter with a diameter of 142 mm. Each filter was subsampled by preserving one quarter in ethanol and freezing another quarter in liquid nitrogen. In 2016 and 2017, 2.5 L Niskin bottles mounted on the ROV captured water above corals. Upon recovery of the ROV, this water was filtered through 0.22 micron filters which were frozen in liquid nitrogen.

16S rRNA metabarcoding

The microbiomes of *Callogorgia delta*, *Callogorgia americana*, water, and sediment were analyzed using 16S metabarcoding. DNA was extracted from *C. delta* tissue and sediment samples using DNeasy PowerSoil kits (Qiagen, Hilden, Germany) following manufacturer protocols using approximately 1 cm of coral branches and about 0.25 g of sediment. DNeasy PowerSoil kits were also used for water samples collected in 2015 using 1 cm² of filter.

Other samples involved different processing methods. DNA was extracted from *C. americana* samples using DNeasy allprep kits with β -mercaptoethanol added to the RLT+ lysis buffer following the manufacturer's instructions. Subsamples from one colony were extracted using both kits. Additionally, subsamples from five *C. delta* colonies were processed using both kits. For all water samples from 2015, replicate extractions were performed on quarters of the filter preserved in both ethanol and frozen in liquid nitrogen. DNA was extracted from all other water samples using Qiagen DNeasy PowerWater kits. See Table S1 in the Supplementary Information for all details regarding processing methods for all samples. Some colonies were sampled multiple times across years and multiple times across the colony. Colonies were identified each year by location and unique branching pattern using images and geographic coordinates from previous collections. ASVs were considered detected in a colony if they were detected in at least one replicate of the colony. Only the first replicate from frozen tissue was used to report relative abundances or in Fig. 1.”

We also added several small changes to the results section to clarify the details of the samples used to calculate each reported number. We also restricted calculations of average relative abundances in *C. delta* to samples from frozen tissue and extracted with powersoil kits.

Results Line 94-123: “We investigated the presence of novel mollicutes using V1-V2 16S rRNA-based metabarcoding on an Illumina MiSeq. Three abundant amplicon sequence variants (ASVs) were identified among 108 *Callogorgia delta* colonies that belong to the class Mollicutes (Molli-1,2,3). Molli-1 was the most prevalent ASV (Fig. 1b). It was detected in 99 out of 108 colonies and was present in samples from all three sampling years (2015-2017) and all six sampling locations (Fig. 1b). It had an average relative abundance of 77% among corals which harbored it (determined from only frozen samples processed with DNeasy powersoil kits = FPS) and reached a maximum of 99% of the microbial community in some samples. It was also present in all four *Callogorgia americana* colonies processed with DNeasy allprep kits averaging 93% of the microbial community. Molli-2 differed from Molli-1 by a single base pair among 300. This closely related ASV was detected in 4 of 10 colonies from the deepest site, GC290. Among those four colonies, Molli-2 averaged 90% (FPS) of the microbial community while Molli-1 was undetected (Fig. 1b).

Molli-3, which was 72% identical to Molli-1, was prevalent and abundant in some colonies, but less so compared to Molli-1 (Fig. 1b). Molli-3 was present in 69 out of 108 *C. delta*

colonies (mean 14.5% when present, up to 82% of the community, FPS) and two out of four *C. americana* colonies (up to 15%). However, Molli-3 was absent in all *C. delta* colonies from the deepest sites GC249 (n=14) and GC290 (n=10).

None of these ASVs were detected in any water sample (n=30). However, Molli-1 and Molli-3 were detected in 35 and 11 out of 44 total sediment samples with maximum relative abundances of 1.1 and 0.16%, respectively. Molli-1 was detected in three of four sequencing blanks.

In addition to novel mollicutes, the most abundant ASVs among *Callogorgia delta* colonies were classified as Thiobarbaceae (mean 7.5% per colony, FPS), Endozoicomonas (1.9%), Thioglobaceae (1.5%, most classified as SUP05 cluster), and Shewanella (1.2%) (Fig. S1). ASVs with other classifications averaged a total of 6.6% and included other common coral associates such as corallicolid apicomplexans and Rickettsiaceae.”

We also added a statement to the end of the legend of Figure 1 to clarify the details of all the samples included in the barplot in panel c.

Fig. 1 Line 1137-1141 “*C. americana* samples were extracted with a DNeasy allprep kit and *C. delta* samples from 2015 were preserved in ethanol. All other samples shown were frozen and extracted with DNeasy powersoil kits. For colonies with multiple replicates, the first replicate using frozen tissue is shown and others excluded.”

Comment 1.5 Lines 82-86: The detection of this Mollicutes in different samples and different years is an interesting data. I would avoid comparing relative abundances, distributions and specificities (lines 86-88) unless the sampling design of the samples at very specific locations and the processing of the samples can be clarified (and I would not compare it with samples processed using different sampling tools and DNA extraction protocols).

Response 1.5 We added details to these lines to clarify that the 77% are derived from samples that are comparable. Please see quoted text from the results section in Comment 1.4 above.

Comment 1.6 Lines 225-231: How and from how many samples was this expression evaluated? How can this be considered high (compared to what and using what statistical methods)?. More details are required regarding methods, standards and goals of using the RNA data.

Response 1.6 The line numbers associated with each comment don't always seem to match the comment. These are the genes mentioned in the results that were among the top 10 most highly expressed genes. We changed this sentence to state this more directly and avoid the confusion associated with simply calling it “high expression”

Discussion Line 314-316: “Its genome contained an extensive CRISPR array and some of the most highly expressed genes were restriction-modification systems and endonucleases.”

Comment 1.7 Lines 324-327: Please provide images and more details of the specially designed tools and containers.

Response 1.7 We included the following to describe the sampling containers.

Methods Line 461-467: “These consisted of “coral quivers” and “bioboxes” composed of high-density polyvinyl chloride (HD-PVC, Fig. S3). The coral quivers consisted of a cylinder (15 cm internal diameter by 25-40 cm tall) that was sealed with a rubber stopper. The bioboxes were sealed with a rubber gasket and contained internal dividers. Biobox dimensions varied between 30.5-65 cm long, 30.5-50 cm wide, and 25-30.5 cm tall and were composed of 1.9 cm thick HD-PVC.”

We do not have permission to display images of all the tools and equipment since they are designed by private companies. We provide a picture of a coral quiver in the Supplementary Information (Fig. S3) and provide links to descriptions and images of the bioboxes and coral cutters.

Comment 1.8 Lines 320-323: I would not keep this comparison due to the very low number of replicates (i.e., n=3 and even n=2 in one of the locations) from sites that seem quite far from each other.

Response 1.8 We noticed that one sample was actually a replicate of a colony so there were only 4 total colonies sampled. We understand the limitations of small sample sizes however we disagree that these samples should be dropped. We did not compare the relative abundance between these two species of *Callogorgia* or between sites. None of our conclusions are based on differences in relative abundance between coral species and across sites. Our goal was to determine if Oceanoplasmataceae could also be detected in *Callogorgia americana*. We detected these mollicutes in all four colonies of this second species and we reported their relative abundances. It would be inappropriate to ignore this finding. The data demonstrates that these novel mollicutes can be found in the related species *C. americana*.

We changed the section of the methods describing the purpose of including these samples to better justify their inclusion and to clarify our goals.

Methods Line 444-449: “To determine if novel mollicutes could also be detected in closely related coral species, we additionally processed four *Callogorgia americana* colonies that were collected in 2010 from Garden Banks (GB) 299 (n=2 colonies, 358-359m, 27.689 N -92.218 W) and VK862 (n=2 colonies, 352-357m, 29.109 N -88.387 W).”

Comment 1.9 Lines 331-337: The samples should be ideally collected using the same tools and strategies. Why were water samples collected using different methods?

Response 1.9 Yes, we agree. It is ideal to sample the same way across multiple years to maximize comparability. We initially used the McLane large volume pump in 2015. We planned to use this in 2016 and 2017 too. In the end, this was not possible. This pump was large and difficult to deploy with an ROV. While recovering it, the ROV manipulator accidentally punched its lens to its camera housing causing irreparable damage that rendered it unusable. Therefore on later cruises, we used niskin bottles to collect water samples instead of the McLane pump. In the end, the additional sampling methods were more relevant for the goal of this manuscript which was to determine if Oceanoplasmataceae could be detected in the water. They were not detected in any water sample of any type.

Comment 1.10 Lines 342-343: If the DNA was extracted using different kits (on top of the different sampling processes/tools) these results are not comparable.

Response 1.10 We understand the limitations of comparing samples processed using different extraction kits and other processes. However, the majority of these samples were frozen and DNA was extracted using DNeasy powersoil kits. This includes all sediment samples and all *Callogorgia delta* samples except those collected in 2015. It also includes the water collected in 2015. We avoided making inappropriate comparisons and conclusions that could be influenced by different methods.

Comment 1.11 Lines 363-366: From which colonies? Collected in the same year, same sites? Please add more details for this entire section.

Response 1.11 We added these details to this sentence and we created Table S2 to provide the collection year and site for all metagenomic, metatranscriptomic, and microscopy samples.

Methods Line 523-524: “DNA was extracted from eight *Callogorgia delta* colonies as described in Vohsen et al. 2020 (collected in 2016, four colonies from MC751, two from MC885, two from GC234, Table S2).”

Comment 1.12 Lines 370-372: Same as above, it is not clear from which colonies, collected where and when, these metagenomes were obtained and why. Although I understand the goal was to assemble the genome, this information still needs to be included.

Response 1.12 See response to Comment 1.11 above. We added a reference to Table S2 where this information can be found.

Comment 1.13 Line 373 and lines 397-403: Although a low number of replicates was acceptable

for RNA analysis a few years ago, I don't think reporting data without a minimum n=3 is still acceptable.

Response 1.13 We agree that there are limitations when using small numbers of transcriptomes. A minimum of 3 transcriptomes per sample group is a general rule when comparing two groups statistically to identify genes that differ in expression levels between those groups. This was not our goal with our transcriptomes. We sequenced RNA to understand which genes across the mollicute genome were expressed and which ones were the most highly expressed. This was invaluable when investigating the identity of predicted genes because it allowed us to filter out genes that were more likely to be artifacts (not detected in the transcriptome) and also allowed us to focus on the predicted genes with the highest expression levels. In this manner we could identify 3 unknown genes that may be important to the functioning of these mollicutes. These transcriptomes also pointed to the importance of the arginine dihydrolase pathway and mechanisms to interact with foreign DNA as they were the most highly expressed pathways in those samples.

We recognize however that additional transcriptomes would strengthen our confidence about which genes are more highly expressed than others. Therefore, instead of dropping these valuable transcriptomes, we now explicitly state that the assessment of which genes were most highly expressed is limited by the small sample size.

Discussion Line 327-329: “However, the metabolic insights gleaned from the most highly expressed genes in this study are limited by a low sample size (n=2) and may not be reflective of general expression levels.”

Comment 1.14 Lines 397-403: What criteria was used for a specific expression to be considered “high”?

Response 1.14 We meant among the top 10 most highly expressed genes. We clarified by making the following change.

Methods Line 581-584: “Coding regions among the 10 most highly expressed but lacking functional annotations from RAST or DRAM were further investigated using blastp and HHBlits⁹³ to inform potential function.”

Comment 1.15 Lines 416-443: The comparative genomic analyses of the obtained genomes should probably be the focus of the paper, as the ecological aspects and comparisons seem rather disconnected at times and also lack a more appropriate experimental design and standardized processing of the samples.

Response 1.15 We agree. The comparative genomics analysis and the microscopy is the greater focus of the manuscript. It is the topic of the bulk of the results and discussion.

Comment 1.16 Lines 443-481: Were Mollicutes previously located in the tissue of other deep-sea octocorals? If not, this is another strength of the paper, which would present the comparative genomic analyses and location of mollicutes, including new representatives, in deep-sea octocorals.

Response 1.16 Mollicutes were not previously located in the tissue of octocorals.

Comment 1.17 The resolution of the figures could be improved, but I am unsure if this is an artifact of the built pdf.

Response 1.17 We believe this is a result of the pdf building process.

Comment 1.18 Overall, a detailed table or figure explaining the sample collection, a map indicating the sampling sites and distances, the number of colonies collected each year and the proximity between colonies collected in different years and how and when environmental samples were collected would be very useful.

Response 1.18 We have created three tables to convey all this information in a much more easily digestible manner in combination with Figure 1. Table S1 displays all the information for the samples in the 16S dataset. Table S2 summarizes which samples were used for metagenomes, metatranscriptomes, and microscopy. Table S3 displays the distances between sites and the maximum distances between colonies within a site.

We also created a supplemental file containing the distances between samples using their coordinates. We included the statement below to the methods to describe distances between corals within sites. The additional tables are placed within the supplement to keep the focus on the genomics analysis of the mollicutes as suggested by the reviewer in **Comment 1.15**.

Methods Line 443-444: “Within a site, corals spanned a range from within a meter apart to a maximum of 32 – 1514 m apart depending on site (Table S3, Supplementary File 1C).”

Reviewer 2:

Comment 2.0: This paper was extremely well-written, easy to read and understand, and the figures beautifully done. The methods are impeccable. The 16S rRNA gene and metagenomic data strongly suggests that these bacteria are abundantly associated with the tissue. The authors were able to construct complete assemblies for these extraordinarily streamlined genomes, allowing for hypotheses about how they interact with the coral hosts.

Response 2.0 Thank you for the supportive comments.

Comment 2.1 However, this paper is significantly weakened by the lack of FISH evidence specific to the new group found here – I think it’s reasonable to hypothesize based on the abundant bacterial populations (as observed via FISH) and the 16S and metagenomics data that these are associated with the tissue, but specific FISH probes would dramatically strengthen the claim. They also do not present TEM, which also seems important to their argument (if it is possible to do for the mesoglea) – for example to show the presence of cells with the characteristic shapes of Mollicutes (pear shaped, spiral shaped).

Response 2.1 We agree that having FISH microscopy using specific probes would strengthen the evidence that Oceanoplasmataceae are tissue associated. We have since tried to replicate our findings with probes that are specific to Oceanoplasma and Thalassoplasma but our samples have proven too old for that purpose. Obtaining new samples would require an ocean-going cruise to the tune of \$40,000/day which takes years to organize. However, existing samples proved suitable for generating novel TEM micrographs. They show that the signals we see from FISH microscopy with general probes are associated with bacteria with the characteristic morphology of mollicutes. We also confirmed their location within the mesoglea showing their presence within the acellular matrix of collagen fibers. We have added TEM to the methods, figures (Fig. 5 and 6), results, and discussion. Also, we now explicitly state in the discussion that using specific probes would strengthen the case that these bacteria are Oceanoplasma.

Methods Line 691-714: “During subsampling, 0.5 cm long segments from the tips of branches were fixed in 2.5% (vol/vol) glutaraldehyde, 9% (wt/vol) sucrose, and 1.5X PHEM buffer (90mM PIPES, 37.5mM HEPES, 15mM EGTA, 3mM MgCl₂) at 4 °C overnight (~8 hours). Samples were then washed with 1.5X PHEM and 9% sucrose thrice and stored in the final wash buffer. Samples were decalcified 2-2.5 weeks later using 0.4M EDTA, 1.5X PHEM, and 9% sucrose for 3 days then washed three times with 1.5X PHEM in 9% sucrose. Samples were then washed twice in 0.1M Na cacodylate buffer and postfixed for 1 hour in 1-2% osmium tetroxide in 0.1M Na cacodylate buffer. Samples were then washed twice in cacodylate buffer and then with milliQ water. Samples were then stained *en bloc* with 2% aqueous uranyl acetate overnight then dehydrated with the following ethanol series: 25%, 50%, 70%, 85%, 90%, 95%, 100%. Next, samples were further dehydrated with three washes of 100% molecular grade ethanol and three acetone washes. Then samples were infiltrated with a 50:50 mix of acetone and Spurr’s epoxy resin for 24 hrs, then 100% Spurr’s epoxy resin for 24 hrs three times. Finally, samples were placed in molds, filled with fresh Spurr’s epoxy resin then baked at 60 °C overnight to polymerize blocks.

Blocks from six *Callogorgia delta* colonies were sectioned on a Leica EM UC6 Ultramicrotome (Leica Microsystems, Wetzlar, Germany) using an Ultra 45° diamond knife (DiATOME, Nidau, Switzerland) to a thickness of 70 nm. Sections were retrieved on 200 mesh

copper grids with formvar and carbon coating and viewed on a JEM-2100 Electron Microscope (JEOL, Tokyo, Japan) with an accelerating voltage of 120kV

To provide an overview of tissue morphology, thick sections (500nm) were stained with epoxy tissue stain (toluidine blue and basic fuchsin) and viewed with a Nikon Eclipse E1000 microscope (Nikon Corporation, Tokyo, Japan).”

Results Line 241-262: “To locate the symbiotic bacteria, we used probes that bind to ribosomal RNA, permitting deposition of a fluorophore through catalyzed reporter deposition fluorescence in situ hybridization (CARD-FISH). Fluorescence representing abundant bacteria was observed in the mesoglea of four separate *Callogorgia delta* colonies (Fig. 5 and 6). These signals were present in tissue sections hybridized with the EUB338 I-III probe mix but absent in adjacent sections hybridized with the negative control probe (non-EUB). These signals were <1 μm in diameter and overlapped DAPI fluorescence. These bacterial signals were observed in both the mesoglea within the coenosarc surrounding the proteinaceous central axis (Fig. 5bcd) as well as the mesoglea within polyps (Fig. 6abc), and in mesoglea within the coenosarc surrounding the proteinaceous central axis (Fig. 5a). They formed large aggregates that conformed to the shape of the mesoglea and potentially contained of hundreds or thousands of cells (Fig. 6b and S4). Another fluorescence signal was observed in the base of the tentacles in one colony but was far less aggregated (Fig. S4).

Transmission electron microscopy confirmed that bacteria resembling *Mollicutes*⁶¹⁻⁶⁴ were present in the mesoglea around the axis in five of six colonies (Fig. 5efg) and in the mesoglea of polyps in four of six colonies (Fig. 6de). The mollicute cells lacked cell walls, were about 0.25 microns wide, and were pleomorphic with irregular shapes that were often elongated and possibly filamentous. These mollicutes were found in the acellular matrix of collagen fibers between tissue layers that forms the mesoglea (Fig. 5g).”

Discussion Line 273-274: “Additional evidence to confirm the presence of *Ca. O. callogorgiae* and *Ca. T. callogorgiae* in the coral mesoglea could come from applying species-specific FISH probes.”

Comment 2.2 The streamlined genome suggests an obligately host-associated lifestyle, which can be achieved through vertical transmission or through individual-to-individual lateral transmission with no free-living environmental stage. What is the model for transmission here given the host is sessile (meaning the potential for direct inter-individual transmission seems low)? And how is that reconciled with the presence of Oceanoplasmataceae in the sediment samples?

Response 2.2 We did not propose a model for transmission however we agree that the highly reduced genome suggests that Oceaplastmataceae is obligately associated with its host. We predict that it is transmitted vertically through larvae as you described or through grazing predators.

Discussion Line 297-304: “It is not clear how these symbionts are transmitted between coral colonies. Their streamlined genomes suggest they do not have a free-living stage. Therefore, they may be transmitted vertically through coral larvae. Additionally, grazing fauna such as snails may serve as vectors transmitting symbionts between coral colonies. In shallow-water corals, corallivores including grazing gastropods are associated with the spread of bacterial and fungal pathogens⁶⁸. Further, other mollicutes infect sessile hosts and are transmitted by mobile vectors, such as phytoplasmas that infect plants and are spread by insects⁶⁹.”

We have added a section to discuss the detection of Oceanoplastmataceae in the sediment but do not propose a role in transmission.

Discussion Line 281-292: “It is possible that Oceanoplastmataceae are abundant in the sediment but only comprise a low relative abundance because the total abundance of the entire microbial community may be high. Another possibility is that DNA from *Callogorgia* and its symbionts can be found in the sediment as environmental DNA originating from sources such as the fecal matter of grazers. Interestingly, eDNA from fish species is orders of magnitude more abundant in the sediment compared to the surrounding water and degrades slower^{66,67}. Alternatively, their detection in sediment libraries may be due to low levels of contamination from *Callogorgia* samples such as through tag jumping or lane jumping. Up to 5 Molli-1 sequences were detected in three out of four sequencing blanks whereas it averaged 19 reads in sediment samples and over 7,000 in *Callogorgia* libraries. See Supplementary Information for further details.”

Comment 2.3 Line 170: The sentence regarding 16S coverage and binning is not clear, please rephrase for clarity.

Response 2.3 We clarified that sentence by changing it to:

Results Line 211-2215: “To do so, we generated a metagenome assembly using the sequence library from that study with the highest coverage of “Mycoplasma” 16S rRNA. We then mapped all four libraries to the metagenome assembly to use for binning with metabat⁵⁹ and identified bin 1 as a member of Oceanoplastmataceae.”

Comment 2.4 Line 209-210: This feels like too strong of a statement to make at this point, that these are specific to Callagorgia, based on a few specimens and the environmental samples present here.

Response 2.4 We changed this line to express less certainty.

Discussion Line 278-280 “Additionally, these two novel mollicute species may associate specifically with corals in the genus *Callogorgia* since both were absent ~~or rare~~ in the surrounding water and rare in sediment as indicated by their low relative abundance but were also detected in the closely related species *C. americana*”

Comment 2.5 Have you searched publicly available 16S datasets for the presence of this group? This can be done reasonably simply with the tool IMNGS.

Response 2.5 Thank you for the suggestion. We have since searched for the presence of Oceanoplasmataceae sequences in other samples using the IMNGS tool. We found Oceanoplasmataceae in additional jellyfish as well as sponges expanding the host range of this family by another host phylum (Porifera). We added a description of our search to the methods, discussed the results, and modified statements in the abstract and discussion.

Abstract Line 17-18: “We erect the novel family Oceanoplasmataceae which includes these symbionts and others associated with five marine invertebrate phyla”

Results Line 136-139: “Additionally, related sequences were identified from publicly available 16S rRNA amplicon datasets using IMNGS⁴⁹. Sponges (*Cinachyrella* and *Suberites*) and jellyfish (*Mastigias*) from marine lakes and open water in Indonesia hosted bacterial communities comprised of up to 86% of these related sequences.”

Discussion Line 392-394: “These hosts are phylogenetically diverse comprising five animal phyla, exhibit diverse lifestyles from being photosynthetic to wood-borers, and originate from a wide range of habitats from the epipelagic zone to hadal trenches”

Methods Line 619-625: “Additional hosts of the family Oceanoplasmataceae were identified in publicly available 16S rRNA amplicon datasets using IMNGS⁴⁹. Seven full-length 16S rRNA gene sequences representing the known breadth of Oceanoplasmataceae were queried against the IMNGS database using the lowest identity threshold (90%). These sequences represented *Ca. Oceanoplasma*, *Ca. Thalassoplasma*, and the symbionts from the five other hosts mentioned previously. Samples with related sequences that totaled more than 5% were considered potential hosts.”

Comment 2.6 Line 263: Do you mean protection from animal viruses? If so, can you use the CRISPR spacers to determine what types of viruses they were interacting with?

Response 2.6 Yes, we meant animal viruses. Identifying the targets of CRISPR spacers is incredibly difficult when they originate from poorly studied hosts and habitats and target undiscovered viruses. We tried blasting these sequences however most result in disparate sets of matches from viruses to terrestrial animal genomes. This is because the sequences are short (30-36bp). To provide some insight into the targets of these spacers, we used the CRISPRCasDB which blasts a query sequence against a database of CRISPR spacers from reference bacterial genomes. This identified three spacers with matches in the database to spacers from marine bacterial genomes. Unfortunately, this did not provide much insight into these spacers beyond this. Still, we added these results to the results section.

Results Line 185-188: “CRISPRCasdb⁵⁴ identified three spacers that resembled those found in the genomes of other marine bacteria (*Empedobacter falsenii*, *Sulfodiicoccus acidiphilus*, and *Nostoc sphaeroides*: all alignment lengths 17/30 bp, all blast e-values 0.007).”

Reviewer 3:

Comment 3.1 The 16S rRNA specific primers used in this study are valid though not commonly used across microbiome studies. One query is the coverage of these primers for the bacterial taxa? Do the primers miss certain groups and hence is the dominance of the Oceanoplasmataceae artificially elevated through primers not targeting certain common octocoral lineages? This information could be added in the supplementary section potentially. The metagenomic approach nicely supports the dominance of the novel Mollicutes clade, though adding further confidence that the 16S profiles are representative of the true octocoral associated microbial community would be advantageous.

Response 3.1 Thank you for your comments. The primers we used (V1+V2: 27F,355R) have been used several times in studies focused on corals and amplify bacterial groups broadly including common octocoral associates such as *Mycoplasma*, *Endozoicomonas*, corallicolids, and Spirochaetes¹⁻⁸. We used these primers because the more common primer pair (V3+V4: 515F,806R) that is suggested by the Earth Microbiome Project amplifies the mitochondrial 12S rRNA gene of many coral species and wastes sequencing effort^{9,10}. The V1+V2 primers amplify common octocoral bacterial lineages in addition to Oceanoplasmataceae (see below) so the dominance of Oceanoplasmataceae is not due to primer bias. We included a description of the other microbes found in *Callogorgia* to the results section and included a figure in the supplement (Fig. S1).

Results Line 118-123 “In addition to novel mollicutes, the most abundant ASVs among *Callogorgia delta* colonies were classified as Thiobarbaceae (mean 7.5% per colony, FPS), Endozoicomonas (1.9%), Thioglobaceae (1.5%, most classified as SUP05 cluster), and Shewanella (1.2%) (Fig. S1). ASVs with other classifications averaged a total of 6.6% and included other common coral associates such as corallicolid apicomplexans and Rickettsiaceae.”

Comment 3.2 The 16S rRNA gene bacterial sequence data primarily focuses on the Mollicutes affiliated sequences. There is no presentation of other bacterial taxa identified from the 16S data, some reference to this other diversity profile of the octocorals would be informative. Are these other sequences affiliated with common marine invertebrate associates as well?

Response 3.2 We included a description of the other microbes found in *Callogorgia* to the results section and included a figure in the supplement. See previous comment (3.1) for text added.

Comment 3.3 Are certain energy sources for the novel Oceanoplasmataceae family derived directly from hydrocarbon seeps, which supports their growth. From the genome reconstruction for example are there metabolic pathways that may indicate metabolism of reduced chemicals such as methane or hydrogen sulfide if the host is found near these seep areas (as stated in the manuscript)? Also though oxygen is low at deep sites is there some level of oxidative pathways supporting growth?

Response 3.3 Reliance of coral symbionts on seep-derived compounds is an intriguing topic. We explore this in other symbionts of other corals in Vohsen et al. 2020⁷. However, here we found no genes or pathways encoded in the genomes that indicate any ability to access any reduced compounds from the seeps. There were also no oxidative pathways to support growth. The only pathway that could generate ATP was the non-oxidative arginine dihydrolase pathway. While *Callogorgia delta* seems to have some kind of association with seeps or their periphery, there doesn't seem to be any link between these novel mollicutes and seeps. This is consistent with the fact that Oceanoplasmataceae also dominated the microbial communities of *Callogorgia americana* which is not found near seeps. We added a statement to the discussion to explicitly state this since we did initially consider a possible connection to seeps.

Discussion Line 330-335: “While *Callogorgia delta* is found near cold seeps, neither *Ca. Oceanoplasma callogorgiae* nor *Ca. Thalassoplasma callogorgia* appear to have an association with seeps. Their genomes did not possess any genes or pathways that could utilize compounds from seeps for energy or nutrition such as reduced sulfur species or hydrocarbons such as light alkanes. Further, both mollicutes were also detected in *C. americana* which is not found near cold seeps.”

Comment 3.4 Overall the data presented is robust and compelling; the only place where some further information or discussion would be valuable is the localization work. Generic Eubacteria probes were used for this visualization and while it is likely this detects the novel Mycoplasma taxa, you can not be certain all the detected signals are from this novel Oceanoplasmataceae group. While 16S and metagenomic information supports that they are the dominant taxa, it is an assumption that is being made and should be qualified. In addition it would be nice to show high

resolution histological images side by side with the FISH images. The resolution of the FISH images are not great and the histological images can help resolve the location of the bacteria in the octocoral tissues. It stated that the bacterial cells are within mesoglea tissue regions though this can not be established from the FISH images. Specific stains (i.e. H&E, Gram, Mallory) can resolve the bacterial localization relative to the host tissues structures. The FISH images then confirm bacteria and localization within the mesogleal tissues layers. Currently just based on the FISH images, there is poor resolution of bacterial cells in relation to host tissues structures.

Response 3.4 Thank you for your comments. We agree that using specific probes would strengthen the case that these signals are Oceanoplasmataceae, that we should be clear that we are assuming these signals are Oceanoplasmataceae since they are the most abundant in the sequence data, and that additional microscopy would help confirm the localization within the mesoglea. We have since obtained TEM micrographs demonstrating the presence of characteristic mollicute cells in the mesoglea. Unfortunately, we could not apply FISH probes with symbiont-specific sequences because the samples we preserved for FISH have degraded to the point where even DAPI staining was no longer detectable. We now present figures 5 and 6 to include TEM micrographs. The new figures still contain some images with poor resolution however we believe their new organization presents the location of these bacteria in the corals more clearly.

Please refer to the response to Comment 2.1 above to see all changes made to the text. In addition to the TEM results and methods, we added a statement to the discussion to qualify the findings and express the possibility that some signals may be due to other bacterial groups.

Comment 3.5 Line 14: provide example of what mechanisms are used (though abstract maybe word limited, if there is capacity additional information here can help the reader).

Response 3.5 We added details about which mechanisms by changing...

Abstract Line 14-17: “Their genomes feature several mechanisms to interact with foreign DNA, including extensive CRISPR arrays and restriction-modification systems, which may indicate their role in symbiosis.”

Comment 3.6 Line 59: Provide a reference to support statement of coevolution of *Mycoplasma* taxa with their host; or detail more explicitly this evidence from the references in the previous sentence.

Response 3.6 Thank you for the comment. We realized that we meant adaptation of *Mycoplasma* to coral hosts as opposed to co-evolution. We modified that line to reflect that change, to be more clear, and to cite the specific study.

Introduction Line 66-69: “Co-occurring coral species host different *Mycoplasma* variants³¹ suggesting that they may have adapted to their coral hosts and closely interact with them as symbionts.”

Comment 3.7 Line 200-203: The image resolution is quite poor for figures 5 and 6, especially in relation to the host tissue structure. Providing histology paired images with the FISH images would help resolved localization. For example, the cells are stated to be in aggregates, though this is hard to visualize in the images. Are these like CAMAs observed previously in scleractinian corals?

Response 3.7 We apologize for the resolution of some images. We have since updated these figures since conducting TEM which provides the resolution and magnification necessary to resolve their localization within the mesoglea. These mollicutes occur in aggregates that appear different in structure to the CAMAs in other corals which are often very dense and spheroid. These mollicutes form aggregates that appear to fill and conform to the morphology of the mesoglea. We added a description of this to the results.

Results Line 252-254: “They formed large aggregates that conformed to the shape of the mesoglea and potentially contained hundreds or thousands of cells (Fig. 6b and S5e).”

Comment 3.8 Line 209-211: However, sequences of these novel taxa were commonly retrieved also from the sediments (as reported in lines 98-100; albeit at low abundance). Hence this statement needs to be refined. Their low abundance retrieval from sediments is partly influenced by the higher diversity of sediment samples; hence not really supported in concluding they are specific to the corals as could be some free-living stage as well. This goes to the previous comment made - are there potential energy sources from seep-environments that also support these novel lineages?

Response 3.8 We could find no genes or pathways in Oceanoplasmataceae genomes that suggest a link to the energy sources at seeps. We have added some text to the discussion to explicitly state this. Please refer to the response to Comment 3.3 above to see the text added.

As you suggest, it is possible that *Oceanoplasma* and *Thalassoplasma* occur in the sediment and are outnumbered by other microbes. We discuss this and other possibilities including the existence of coral and symbiont eDNA in the sediment and contamination. We include further details about the detection of Molli-1 in sediment and sequencing blanks in the Supplementary Information.

Discussion Line 281-292: “It is possible that Oceanoplasmataceae are abundant in the sediment but only comprise a low relative abundance because the total abundance of the entire microbial

community may be high. Another possibility is that DNA from *Callogorgia* and its symbionts can be found in the sediment as environmental DNA originating from sources such as the fecal matter of grazers. Interestingly, eDNA from fish species is orders of magnitude more abundant in the sediment compared to the surrounding water and degrades slower^{66,67}. Alternatively, their detection in sediment libraries may be due to low levels of contamination from *Callogorgia* samples such as through tag jumping or lane jumping. Up to 5 Molli-1 sequences were detected in three out of four sequencing blanks whereas they averaged 19 reads in sediment samples and over 7,000 in *Callogorgia* libraries (Supplementary Information).”

Comment 3.9 Line 216: Not clear what is meant by “genomic coverage of the symbiont was up to 50 times as high as that of the host”. Is that obtained reads were 50x more than the host? That seems unlikely unless some form of enrichment performed? Statement is unclear.

Response 3.9 Sorry for the confusion. We meant to say that the average depth of coverage across the *Oceanoplasma* genome was up to 50 times greater than the average depth of coverage of contigs originating from the coral’s genome. For example, in the library used to assemble the genome of *Oceanoplasma*, the mean depth of coverage across *Callogorgia*’s genome was estimated to be 102 whereas the mean depth of coverage of *Oceanoplasma*’s genome was 3,545 (33 times higher). We did not specifically enrich for bacterial DNA in these tissue extractions but instead applied a standard DNA tissue protocol. It is remarkable how abundant these mollicutes are in these corals. To minimize confusion, we removed this part of the sentence.

Discussion Line 269-273: “Finally, *Ca. Oceanoplasma callogorgiae* likely resides within the coral. Abundant bacteria were observed in the mesoglea of *C. delta* and *Ca. O. callogorgiae* was the most abundant bacterium by far in both the amplicon dataset and the metagenomes where the genomic coverage of the symbiont was up to 50 times as high as that of the host.”

Comment 3.10 Note: line 222 is repeating this statement “likely receive this arginine from their coral hosts” that is also stated above (lines 211-213).

Response 3.10 We simplified the earlier statement and left details in the later statement.

Earlier Discussion Line 268-269: “Further, their genomes were very reduced and demonstrated a reliance on compounds like arginine and amino acids that are likely obtained from *C. delta*”

Later Discussion Line 310-311: “They likely receive this arginine from their coral hosts as well as essential compounds such as amino acids, riboflavin, and biotin.”

Comment 3.11 Line 231: why if host associated (and in aggregates) do they need these CRISPR systems? Does this indicate a potential free-living life stage (despite reduced genomes)? Would

high levels of viral challenge be expected if residing solely in host tissues. Acknowledge sponge symbionts are enriched in CRISPR-Cas systems (line 284) but this makes sense since large amounts of water are passing through sponges and encounter rates with viruses and other microbes would be high in a sponge, but not in octocoral tissues.

Response 3.11 It is not clear why these novel mollicutes would have CRISPR systems despite being within the mesoglea of *Callogorgia*. Many mollicutes that are host associated encounter mollicutes-specific phages and thus have CRISPR Cas systems. Bacteria residing in the mesoglea may occasionally experience higher environmental exposure such as when the coral is grazed upon or when branches break. We observed several grazing predators on this species and documented that this species suffers much higher levels of branch loss than others in the Gulf. We added this to the discussion.

Discussion Line 370-385: “Others have suggested that the symbionts of hadal sea cucumbers, now identified as belonging to Oceanoplasmataceae, may provide protection to their hosts from animal viruses using their CRISPR-Cas and restriction-modification systems²⁴. It is unclear if this is a possibility for *Ca. O. callogorgiae* since it likely resides in the mesoglea where direct exposure to viruses may be low. Virus-like particles have been observed in the mesoglea of a shallow-water, stony coral⁸¹, however they were rare and far less abundant than those found in the epidermis and gastrodermis. Exposure to viruses in the mesoglea may increase when predators graze on corals or when branches break. Indeed, gastropods have been observed grazing on *Callogorgia delta* colonies and they suffer higher rates of branch loss compared to other deep-sea corals in the genus *Paramuricea* which do not host Oceanoplasmataceae⁸². Further, upon injury, phagocytic amoebocytes migrate from the mesoglea to the wound where they provide defense against invading pathogens⁸³. If *Ca. O. callogorgiae* has a role in immune defense, it may also reside in the mesoglea and migrate when tissue is wounded.”

Comment 3.12 Line 262: Though as stated previously (line 74); they reside near hydrocarbon seeps which can be the predominant source of nutrients?

Response 3.12 There are no genes or pathways in the genomes of these novel mollicutes that suggest any ability to utilize any compounds originating from the seeps. We added text to the discussion to state this explicitly. Please see the response to Comment 3.3 above to see the added text.

Comment 3.13 Line 327: were samples also preserved in fixative’s for FISH work?

Response 3.13 Yes, samples were also preserved in fixatives for both FISH microscopy and EM. We added the following line to the methods. Details about the fixatives can be found in the sections describing the microscopy methods.

Methods Line 467-470: “Upon recovery, corals were maintained at 4 °C for up to 4 hours until subsamples were flash frozen in liquid nitrogen and fixed for microscopy (see below for details).”

Comment 3.14 Line 342: Can any more detail be provided on where on the colonies these branches were taken? i.e. Tips of branches?

Response 3.14 Yes. We sampled the ends of branches and the pieces were around 30 cm in length. The tips of branches were preserved for microscopy. We added these details to the general collection methods and those for processing microscopy samples.

Methods Line 460-4461: “Around 30 cm of the distal end of branches were removed, placed in separate temperature-insulated containers until recovery of the ROV.”

Methods Line 640-642: “During subsampling, 0.5 cm long segments from tips of branches were fixed in 2 mL of 4% formaldehyde with 1X phosphate-buffered saline (PBS) at 4 °C overnight (4-8 hours).”

Methods Line 691-693: “During subsampling, 0.5 cm long segments from the tips of branches were fixed in 2.5% (vol/vol) glutaraldehyde, 9% (wt/vol) sucrose, and 1.5X PHEM buffer (90mM PIPES, 37.5mM HEPES, 15mM EGTA, 3mM MgCl₂) at 4 °C overnight (~8 hours).”

Comment 3.15 Line 348: What is the coverage of these primers? This region of the 16S is less used than others for metabarcoding studies, hence some idea of coverage would be useful to demonstrate that major taxa clades are adequately covered.

Response 3.15 Please see response to comment 3.1 above.

Comment 3.16 Figure 1: The bathymetric map is not a clear way to present the sampling sites i.e. contours are not informative and would suggest a stylized drawn B&W image with the same information of spatial location, site name, depth and number of sample taken from each site would be a better representation.

Response 3.16 Thank you for your input. We have updated the map in figure one. We have changed it from color to gray scale. Instead of shading the bathymetry by depth, we included 500m isobaths.

Comment 3.17 Note in figure legend – states a star signifies water sampling; though do not see the stars.

Response 3.17 The stars appeared on the number of water samples. We removed the star and listed the niskin and McLane pump water samples separately. We changed the figure legend to state this so it is clear and easier to spot. See the new legend for figure 1 below. The highlighted sections show what was changed.

Figure 1 legend: “a) Map of locations where *Callogorgia delta* (black circles) and *C. americana* (gray diamonds) were sampled. Lines represent 500m isobaths. The depth range of samples collected from each site is denoted below the site name followed by the number of colonies, sediment samples, niskin water samples (~2.5L), then McLane pump water samples (~400L) in parentheses. If no water or sediment samples were collected, only the number of colonies is shown. If no McLane pump water samples were taken, only the number of niskin water samples are shown. A star signifies whether a water sample was collected using a McLane pump (~400L) in addition to niskin bottles (~2.5L).”

Comment 3.18 Figure 1: Why not provide an image of *Callogorgia americana* as well?

Response 3.18 We do not possess a high-quality image of this species.

Comment 3.19 Figure 1; highlights that the 16S libraries of some samples are not dominated by the *Oceanoplasma* taxa. Hence what are the profiles of these samples and more broadly what other sequences were retrieved across all samples. See earlier comment on documenting at least briefly the extent of the diversity of these octocorals samples. Accept not the focus of the study but the *Oceanoplasma* seem not to be universally dominant in all samples (though most).

Response 3.19 We have added a description of the other microbes that were detected in *Callogorgia* to the results section. We also added a figure to the supplement to show this. Please refer to response to Comment 3.1 for more details.

Comment 3.20 Figure 5 and 6 could be consolidated easily into one Figure, as there is redundancy in many of the images for the specific details that are informative. See earlier comment though that good histological images would resolve the host tissue structure and microbiome localization much more clearly. For example, good histological staining will identify the mesoglea unambiguously and refine the polyp structures such as mesenteries and tissue layers (endoderm and gastroderm).

Response 3.20 We recognize the redundancy of some images. We have changed these figures since obtaining TEM images. We retained two figures. The first shows where the mollicutes can be found in the mesoglea around the axis whereas the second figure shows them in the polyps. Unfortunately, we could not generate good quality histological images. However, with TEM we were able to confirm the location of mollicutes in the mesoglea supported by the presence of collagen fibers.

References within Responses:

- 1 Cook, G. *et al.* A comparison of culture-dependent and culture-independent techniques used to characterize bacterial communities on healthy and white plague-diseased corals of the *Montastraea annularis* species complex. *Coral Reefs* **32**, 375-388 (2013).
- 2 Rodriguez-Lanetty, M., Granados-Cifuentes, C., Barberan, A., Bellantuono, A. J. & Bastidas, C. Ecological Inferences from a deep screening of the Complex Bacterial Consortia associated with the coral, *Porites astreoides*. *Mol. Ecol.* **22**, 4349-4362 (2013).
- 3 Voss, J. D., Mills, D. K., Myers, J. L., Remily, E. R. & Richardson, L. L. Black band disease microbial community variation on corals in three regions of the wider Caribbean. *Microb. Ecol.* **54**, 730-739 (2007).
- 4 Dannenberg, R. P. Characterization and oil response of the deep sea coral-associated microbiome. (2015).
- 5 Vohsen, S. A. *et al.* Deep-sea corals provide new insight into the ecology, evolution, and the role of plastids in widespread apicomplexan symbionts of anthozoans. *Microbiome* **8**, 34 (2020).
- 6 Vohsen, S. A. & Herrera, S. A seascape approach to the microbial biogeography of mesophotic and deep-sea octocorals. *bioRxiv*, 2023.2007.2005.547877 (2023).
- 7 Vohsen, S. A. *et al.* Deep-sea corals near cold seeps associate with chemoautotrophic bacteria that are related to the symbionts of cold seep and hydrothermal vent mussels. *bioRxiv*, 2020.2002.2027.968453 (2020).
- 8 Sekar, R., Mills, D. K., Remily, E. R., Voss, J. D. & Richardson, L. L. Microbial communities in the surface mucopolysaccharide layer and the black band microbial mat of black band-diseased *Siderastrea siderea*. *Appl. Environ. Microbiol.* **72**, 5963-5973 (2006).
- 9 Pollock, F. J. *et al.* Coral-associated bacteria demonstrate phylosymbiosis and cophylogeny. *Nat. Commun* **9**, 4921 (2018).
- 10 van de Water, J. A. J. M., Coppari, M., Enrichetti, F., Ferrier-Pagès, C. & Bo, M. Local conditions influence the prokaryotic communities associated with the mesophotic black coral *Antipathella subpinnata*. *Front. Microbiol.* **11** (2020).

We thank the editor and all reviewers for the time and effort they have put into reviewing and improving our manuscript. We revised statements to be more transparent and to avoid overstatements. All reviewer comments are presented below in blue text. They have been numbered and are preceded by “Comment x.y” where x is the reviewer and y is their yth comment. Author responses follow each comment in black text preceded by the label “Response x.y”. Text from the manuscript is presented in quotes and underlined. Differences compared to the first submission are highlighted.

REVIEWER 1:

Comment 1.1 The use of varied methods and sample collections across different times and conditions, coupled with the insufficient number of replicates, especially for the RNA analysis, raises substantial concerns about the consistency and reliability of the data. While I understand that differently processed samples can be employed for detecting specific groups, the potential biases introduced by these methodological inconsistencies are not adequately acknowledged.

Response 1.1 We agree with the reviewer that different fixation and DNA extraction methods can lead to variable results. We were fortunate in our study that we had consistent results for the detection of the Oceanoplasmataceae symbionts. To address the reviewer's concerns, we revised our results section on the 16S amplicon data to make the variation in methods clearer and to emphasize that all methods consistently detected the symbionts.

Results: “We investigated the presence of novel mollicutes in *Callogorgia* colonies using V1-V2 16S rRNA-based metabarcoding on an Illumina MiSeq. Colonies were collected from seven locations (Fig. 1a) across four years (2010, 2015-2017). Three abundant amplicon sequence variants (ASVs) were identified among 108 *Callogorgia delta* colonies that belong to the class Mollicutes (Molli-1,2,3). Molli-1 was the most prevalent ASV (Fig. 1b). It was detected in 99 out of 108 colonies and was present in samples from all three sampling years (2015-2017) and all six sampling locations as well as every year each site was sampled (Fig. 1b). Molli-1 had an average relative abundance of 77% among corals which harbored it (determined from only frozen samples processed with DNeasy powersoil kits (=FPS) and reached a maximum of 99% of the microbial community in some samples. It was also present in all four *Callogorgia americana* colonies processed with DNeasy allprep kits averaging 93% of the microbial community. Molli-1 constituted a high proportion of the microbiome in most samples including samples from 2015 which were preserved in ethanol as well as *C. americana* samples which were processed using a DNeasy Allprep kit (Fig. 1b).”

Abstract: “Here, we discovered two mollicutes that dominate the microbiome of the deep-sea octocoral, *Callogorgia delta*, and likely reside in the mesoglea. These symbionts were abundant across the host’s range, absent in the water, and appear to be rare in sediments.”

Introduction: “We ~~assessed~~ ~~determined~~ the specificity of the association to corals by screening sediment and water in addition to *Callogorgia*”

Discussion: “The data presented here suggests that *Ca. Oceanoplasma callogorgiae* and *Ca. Thalassoplasma callogorgiae* are symbionts of *Callogorgia delta sensu Goff*¹. First, they were ~~found~~ ~~detected~~ in *C. delta* colonies from all sites and sampling years.”

Discussion: “The two novel mollicute species may associate specifically with corals in the genus *Callogorgia* since both were absent in the surrounding water and ~~appear to be~~ rare in sediment as indicated by their low relative abundance. ~~However~~, it is possible that Oceanoplasmataceae ~~have~~ ~~a relatively high absolute abundance~~ ~~are abundant~~ in the sediment but only comprise a low relative abundance because the total abundance of the entire microbial community may be high.”

Comment 1.2 The use of unreplicated RNA data to indicate "high expression" still needs clearer justification, as typically, such claims should be supported by statistical analysis against controls. If the intent was to indicate mere "activity" or presence of gene expression, this should be explicitly clarified to avoid misleading interpretations.

Response 1.2 We thank the reviewer for pushing us for higher precision and first want to point out that we had two replicates for the RNA data. Producing additional RNA libraries was not possible for this manuscript due to the inaccessibility of deep-sea habitats. We agree that our RNA data based on two samples is not well-replicated, but serves the purpose to detect transcription and to rank genes by their transcription levels. To clarify, we ranked the transcript abundances across genes to identify which genes were among those with the highest transcript levels. Thus, high transcript levels here are in reference to other genes in the genome. This serves to identify pathways which were highly transcribed and thus potentially underlying important activities of these bacteria. We found that the 8 genes with the highest transcript levels on average were outliers compared to the remainder of genes. Further, the genes whose transcript levels we reference in the text were among the top 5% of genes identified in both libraries. We added text to the introduction to clarify the purpose of the RNA libraries and adjusted the language regarding the RNA data to always give specific details about what is meant regarding high transcript levels. See the following changes. Further, the discussion contains a statement that explicitly warns that the transcript levels observed in these two libraries may not be reflective of expression patterns generally because of the low sample size.

Introduction: “We ~~sequenced metatranscriptomes to complement metabolic inferences by~~ ~~identifying active genes and those with the highest transcription levels.~~”

Results: “Genes from this pathway were represented by transcripts in in both libraries, were among the top fifteen genes with the highest transcription levels (mean transcripts per million (tpm) 9,300 – 96,900) These genes were among the most highly expressed genes in both RNA libraries (Fig. 3ab), and constitute the sole pathway in this genome capable of producing ATP (Fig. 3).”

Results: “One of the five most highly expressed genes in both libraries was t The specificity subunit S of its type I restriction modification system had the 4th highest average transcription level among all genes (mean tpm = 87,800, Fig. 3b).”

Results: “Four predicted genes with no functional annotations were represented by transcripts in both libraries and among the ten genes with the highest transcript levels (mean tpm = 14,500 – 98,500, Fig. 3b).”

Results: “Another of these highly expressed, unannotated gene (U2), was predicted to contain a GIY-YIG endonuclease domain with high confidence (>95%, Hhblits).”

Discussion: “Its genome contained an extensive CRISPR array and some of the most highly expressed genes abundant transcripts in both of our libraries were restriction-modification systems and endonucleases.”

Discussion: “Other, not yet well-understood processes likely underly the basic functioning of these novel mollicutes since some of the most highly expressed transcribed genes in this study had unknown functions.”

Discussion: “However, the metabolic insights gleaned from the most highly expressed genes abundant transcripts in this study are limited by a low sample size (n=2) and may not be reflective of general expression levels across populations.”

Methods: “To determine which genes were expressed transcribed and to quantify their expression transcription levels, the two RNA sequence libraries were pseudoaligned to the gene annotations of both genomes with kallisto ver0.44.0103 using 100 bootstrap replicates, an estimated fragment length of 180 bp with a standard deviation of 20 bp, and an index with a k-

mer size of 31 bp. The transcription levels of all genes were ranked to identify the genes with the highest transcription levels under the conditions present during sampling by sorting by mean transcripts per million (tpm). Among genes with non-zero mean transcription levels, the top eight most abundant transcripts were outliers after log-transformation (greater than the third quartile plus 1.5 times the interquartile range). The top five most abundant transcripts were outliers in each library individually. Coding regions among the 10 most highly expressed transcribed but lacking functional annotations from RAST or DRAM were further investigated using blastp and HHBlits¹⁰⁴ to inform potential function.

Figure 3 Legend: “Expression Transcription levels of *Ca. O. callogorgiae* genes from two libraries sorted by functional annotations.”

Comment 1.3 The manuscript lacks critical FISH evidence needed to confirm the symbiotic interactions of the bacteria within coral tissues. Such visualization is essential to support some of the most important and ambitious claims of the manuscript.

Response 1.3 We agree that additional FISH evidence using probes specific to Oceanoplasmataceae would reinforce this study. However, we provide strong evidence of their stable association within the corals through FISH with general probes, TEM, 16S sequencing, metagenomics, and metatranscriptomics. Long-term and stable associations are the key definition for a symbiotic interaction that we consistently follow with our line of arguments. To address the reviewer's concerns, we have made the change in the abstract below to communicate the uncertainty that these mollicutes reside in the mesoglea. All other data support a symbiotic interaction and our statements effectively communicate this specific uncertainty on the location of the interaction. Further, we provide a statement in the discussion that explicitly states that FISH using specific probes is necessary to further support the claim that these bacteria reside in the mesoglea.

Abstract: “Here, we discovered two mollicutes that dominate the microbiome of the deep-sea octocoral, *Callogorgia delta*, and likely reside in the mesoglea. These symbionts were abundant across the host’s range, absent in the water, and appear rare in sediments.”

Discussion: “Additional evidence to confirm the presence of *Ca. O. callogorgiae* and *Ca. T. callogorgiae* in the coral mesoglea could come from applying species-specific FISH probes.”

Comment 1.4 Additionally, I agree with the other reviewers that the ecological interpretations drawn from 16S sequencing data need further support, considering the limitations of these short

sequences, the variability in sample processing and potential underestimation of the presence of Oceanoplasmataceae in sediments and the brief claims on their transmission - and I am not sure the revised text has fully addressed these concerns. The discussion regarding the prevalence and ecological roles of these mollicutes seem a bit overstated given the evidence provided. A more cautious approach in discussing these findings is advised, particularly in light of the methodological issues noted above.

Response 1.4 We also agree that we should be cautious about the ecological role of the symbionts and not overstate their importance. We largely refrain from making statements about the ecological roles of the mollicutes and explicitly discuss in the discussion that the costs or benefits of hosting *Oceanoplasma callogorgiae* remain unclear. Whenever we discussed results regarding a potential ecological role of the symbiont, we made sure to communicate uncertainty in our statements and mention several possibilities. We also made several additional changes to more clearly communicate uncertainty in our interpretations.

Discussion: “The two novel mollicute species **may** associate specifically with corals in the genus *Callogorgia* since both were absent in the surrounding water and **may be** rare in sediment as indicated by their low relative abundance. However, it is **possible** that Oceanoplasmataceae are abundant in the sediment but only comprise a low relative abundance because the total abundance of the entire microbial community may be high. Another **possibility** is that DNA from *Callogorgia* and its symbionts can be found in the sediment as environmental DNA originating from sources such as the fecal matter of grazers. Interestingly, eDNA from fish species is orders of magnitude more abundant in the sediment compared to the surrounding water and degrades slower^{2,3}. Alternatively, their detection in sediment libraries **may be** due to low levels of contamination from *Callogorgia* samples such as through tag jumping or lane jumping. Up to 5 Molli-1 sequences were detected in three out of four sequencing blanks whereas it averaged 19 reads in sediment samples and over 7,000 in *Callogorgia* libraries. **Further experimental work is needed to confirm the prevalence of Oceanoplasmataceae in the sediment.**”

Discussion: “It is not clear how these symbionts are transmitted between coral colonies. Their streamlined genomes suggest they do not have a free-living stage. Therefore, **it is possible that they are they may be** transmitted vertically through coral larvae.”

Discussion: “Further, it shared additional genomic features with *Ca. Thalassoplasma callogorgiae* including *comEC* to import foreign DNA and enrichment of thymidine kinase genes **that may to** salvage thymidine from DNA degradation.”

Discussion: “It is still unclear if *Ca. Oceanoplasma callogorgiae* incurs a cost or provides any benefit to its coral host. These novel mollicutes may simply be commensals or parasites with a minimal impact on their host corals since they did not display any obvious pathology and nearly every colony appears to be infected.”

REVIEWER 2:

The authors did a great job of addressing my comments and suggestions. I'm especially appreciative of the new TEM data, which I think adds to their evidence.

We thank the reviewer for their positive and supportive comment and the constructive feedback throughout the process.

My remaining comments are minor:

Comment 2.1 Lines 136-138: Clarify what threshold you used for IMNGS? You have in Methods but might be good for readers to also see that here to give context.

Response 2.1 Thanks for pointing this out, we made the following change:

“Additionally, related sequences (>90% identical) were identified from publicly available 16S rRNA amplicon datasets using IMNGS⁴”

Comment 2.2 Lines 185-188: This information is not clear as to why it's relevant. Why is finding spacers also present in other bacteria relevant here? Instead, wouldn't it be more interesting to see what phages these spacers target? Especially given later discussion regarding whether these bacteria could play a role in viral defense for their coral host.

Response 2.2 Thanks for touching on this. We agree it would be far more interesting and informative to determine which phages the spacers target. However, this is incredibly difficult to achieve. None of the other bacteria we detected the spacers in, or the potential phage targets of the spacers, are well-studied and these spacer sequences are very short. We tried using database searches and blasting spacer sequences to find matches and this was the most information we could glean. We added the following sentence to provide more context to the relevance of the database match.

“The potential targets of these spacers could not be identified. However, CRISPRCasdb⁵ identified three spacers that partially resembled those found in the genomes of other marine bacteria (*Empedobacter falsenii*, *Sulfodiicoccus acidiphilus*, and *Nostoc sphaeroides*: all alignment lengths 17/30 bp, all blast e-values 0.007).

Comment 2.3 Line 213: Do you mean highest coverage of 16S rRNA in amplicon analysis of same sample? Or do you mean from the metagenome? Please clarify.

Response 2.3 Thanks for pointing to this detail that needs clarification. We meant highest coverage of the 16S rRNA gene using reads from metagenomes. We made the following change to make this clearer.

“To do so, we ~~generated a metagenome~~ assembled the metagenomic sequence library from that study with the highest coverage of “Mycoplasma” 16S rRNA. We then mapped all four metagenome libraries to this metagenome assembly to use for binning with metabat⁶ and identified bin 1 as a member of Oceanoplasmataceae.”

REVIEWER 3:

Comment 3.1 The authors have addressed all concerns raised through the review process and therefore have no additional comments.

This is a nice study and well deserving of publication in Nature Communications

Response 3.1 Thank you for your supportive comments.

Reviewer #1 (Remarks to the Author):

I appreciate the authors effort to improve the manuscript and be accurate regarding the conclusions. The new version is indeed much better and mostly reflects the obtained data. I still disagree the use of duplicated samples is acceptable, and I believe the authors also agree with me, considering their response "Further, the discussion contains a statement that explicitly warns that the transcript levels observed in these two libraries may not be reflective of expression patterns generally because of the low sample size.". Although I understand the goal and reasoning, the levels are still questionable as they can be random/biased, which is why a minimal n of 3 is required to ensure reproducibility. It would be probably better to exclude this part, in my opinion.

Response:

We understand the concern with the low sample size of transcriptomes. We amended our statement in the discussion to explicitly state that the low sample size precludes statistical analyses. "However, the metabolic insights gleaned from the most abundant transcripts in this study are limited by a low sample size (n=2) which precludes standard statistical analyses and may not be reflective of general expression levels across populations."

We also added a statement in the results following the first mention of transcriptomic data to be more transparent. "Statistical analyses were not conducted to support the ranking of transcription levels due to the low sample size (n=2)."